# TimeFilter: Patch-Specific Spatial-Temporal Graph Filtration for Time Series Forecasting

**Yifan Hu** [1] [*]  **Guibin Zhang** [2] [*]  **Peiyuan Liu** [1] [*]  **Disen Lan** [3]  **Naiqi Li** [1]
**Dawei Cheng** [4]  **Tao Dai** [5]  **Shu-Tao Xia** [1]  **Shirui Pan** [6]

## Abstract

Time series forecasting methods generally fall into two main categories: Channel Independent (CI) and Channel Dependent (CD) strategies. While CI overlooks important covariate relationships, CD captures all dependencies without distinction, introducing noise and reducing generalization. Recent advances in Channel Clustering (CC) aim to refine dependency modeling by grouping channels with similar characteristics and applying tailored modeling techniques. However, coarse-grained clustering struggles to capture complex, time-varying interactions effectively. To address these challenges, we propose TimeFilter, a GNN-based framework for adaptive and fine-grained dependency modeling. After constructing the graph from the input sequence, TimeFilter refines the learned spatial-temporal dependencies by filtering out irrelevant correlations while preserving the most critical ones in a patch-specific manner. Extensive experiments on 13 real-world datasets from diverse application domains demonstrate the state-of-the-art performance of TimeFilter. The code is available at `https://github.com/TROUBADOUR000/TimeFilter`.

## 1. Introduction

Multivariate time series forecasting is a challenging task due to the need to capture both temporal dynamics and inter-channel dependencies. This problem arises in a wide range of real-world applications, including weather forecast-

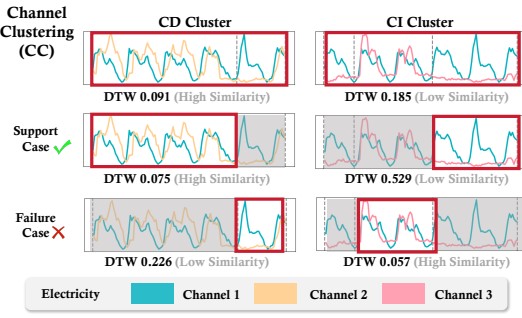

*Figure 1.* Analysis of three channels from the Electricity dataset shows the pros and cons of CI, CD, and CC strategies in different cases. Dynamic Time Warping (DTW) is a metric for measuring the similarity between two sequences, with lower values indicating higher similarity. The CI strategy ignores the highly correlated covariate information in the Supporting Case (left). The CD strategy fuses all information, including the irrelevant dependencies in the Supporting Case (right). The CC strategy addresses these issues by using global correlation to form CD and CI clusters. However, it cannot model the dynamic interactions of channels at different time steps at a fine-grained level. For example, in the Failure Case, it still faces the same dilemma as CD and CI strategies.

ing (Volkovs et al., 2024), traffic flow analysis (Shu et al., 2021), health monitoring (Morid et al., 2023), and financial investment (Zhu et al., 2024a;b). The literature on deep learning approaches for this task can be broadly categorized into channel-independent (CI) methods and channel-dependent (CD) methods (Han et al., 2024b).

CI methods rely on the individual historical values of each channel for prediction (Nie et al., 2023; Dai et al., 2024; Zeng et al., 2023), while CD methods go a step further by modeling the dependencies between different channels (Yu et al., 2024; Huang et al., 2023; Zhang & Yan, 2023). However, it is important to recognize that data from different domains often exhibit significant variations in underlying distributions and characteristics. For instance, in climate-related data, there are typically natural physical dependencies among variables (Karevan & Suykens, 2020), whereas in electricity consumption data, the usage patterns of individual users can vary drastically, with little to no interdependence. This stark contrast suggests that the assumption

---
[*]Equal contribution [1]Tsinghua Shenzhen International Graduate School, Tsinghua University [2]National University of Singapore [3]Fudan University [4]Tongji University [5]Shenzhen University [6]Griffith University. Correspondence to: Tao Dai <daitao.edu@gmail.com>, Shirui Pan <s.pan@griffith.edu.au>.

*Proceedings of the 42nd International Conference on Machine Learning*, Vancouver, Canada. PMLR 267, 2025. Copyright 2025 by the author(s).

of channel independence in CI methods, as well as the indiscriminate modeling of inter-channel dependencies in CD methods, both have inherent limitations.

To address this challenge, recent works have introduced the concept of Channel Clustering (CC), where channels with similar patterns are grouped into clusters based on data-dependent criteria. For instance, CCM (Chen et al., 2024) dynamically clusters channels with inherent similarities, applying CD modeling within each cluster and CI modeling across different clusters. However, CCM only considers relationships within the same cluster. DUET (Qiu et al., 2024b), on the other hand, extends this idea by introducing clustering in the temporal dimension as well. By dynamically assigning different modeling strategies for inter-cluster and intra-cluster relationships, these approaches enhance forecasting capabilities by effectively capturing both intra-cluster and inter-cluster dependencies.

Despite their success, it is notable that existing CC methods typically rely on a coarse-grained approach, where similarity across channels is calculated using data from all time points for clustering. As illustrated in Figure 1, this coarse-grained clustering fails to flexibly select appropriate dependency modeling strategies for specific time periods. The complex dependencies between channels evolve over time, and channels that are closely related within a cluster may become completely independent at certain time intervals. In such cases, applying a CD method could lead to overfitting (Zhao & Shen, 2024). Conversely, channels that are unrelated in the overall cluster may become highly correlated at exact times, and using a CI method would risk losing valuable inter-variable information. This limitation underscores the need for more dynamic and adaptive approaches that can capture time-varying dependencies.

Motivated by the observations above, we transition from previous *coarse-grained, channel-wise clustering* approaches to a *finer-grained, patch-wise partitioning* strategy, where each channel is divided into non-overlapping segments. With this partitioning, as shown in Figure 2(b), there are three key types of relationships: ❶ **temporal dependencies** between different patches within the same channel, ❷ **inter-channel (spatial) dependencies** at the same time step, and ❸ **spatial-temporal dependencies** between different channels and time periods. These relationships are often intricately intertwined in real-world time series data. To more robustly model these complex dependencies, we further design a filtering method for each dependency region, effectively removing irrelevant noisy relationships and ensuring that only the most significant connections are considered.

Technically, we propose TimeFilter, a novel framework crafted to capture intricate spatiotemporal dependencies across both temporal and spatial dimensions, while dynamically filtering out extraneous information. Specifically,

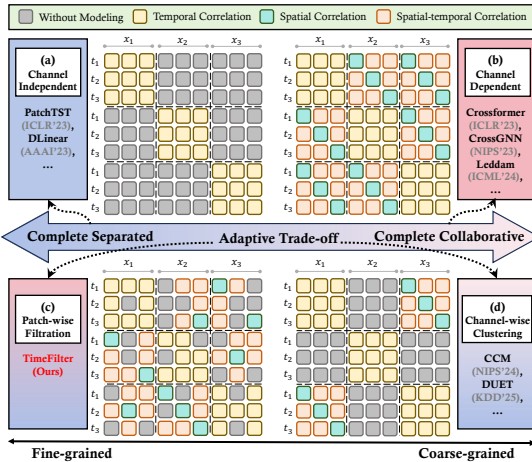

*Figure 2.* The dependency map of 4 different strategies. $t_i$ is the time step and $x_i$ is one channel. (a) CI strategy preserves only the temporal dependencies. (b) CD strategy fuses all dependencies. (c) Patch-wise Filtration finely selects dependencies for each patch. (d) Channel-wise Clustering coarsely models channel dependencies.

TimeFilter begins with **Spatial-Temporal Construction Module**, which segments the input time series into non-overlapping patches and constructs a spatial-temporal graph. After that, **Patch-Specific Filtration Module** applies a Mixture of Experts (MoE)-style dynamic router to filter out redundant dependencies, honing in on the most critical spatiotemporal relationships. Finally, **Adaptive Graph Learning Module** leverages the graph structure to aggregate relevant information and produce accurate forecasting results. Extensive experiments on several datasets show that TimeFilter achieves consistent state-of-the-art performance in both long- and short-term forecasting. Our contributions can be summarized as follows:

❶ *Novel Paradigm.* We *for the first time* advocate for a fine-grained segmentation of dependencies, capturing spatial-temporal relationships at the patch level while adaptively filtering out irrelevant information.

❷ *Technical Contribution.* We propose TimeFilter, a novel dependency modeling strategy that employs patch-specific filtration to learn better omni-series representations for time series forecasting.

❸ *Holistic Evaluation.* Comprehensive experiments demonstrate that TimeFilter achieves state-of-the-art performance in both long- and short-term forecasting, with error reductions of $4.48\%$ and $5.34\%$, respectively.

## 2. Related Work

### 2.1. Multivariate Time Series Forecasting

Multivariate time series forecasting (MTSF) has widespread applications in various domains. In recent years, deep learn-

ing models have shown great promise in MTSF. RNN-based methods (Lin et al., 2023), leveraging the Markov assumption, model the sequential nature by considering current and previous time steps. CNN-based methods (Dai et al., 2024; Wu et al., 2023) apply convolution operations along the time dimension to capture local temporal patterns. However, they struggle with long-term dependencies due to vanishing gradients and limited receptive fields, respectively. Transformer-based methods (Liu et al., 2024b;a) use attention mechanisms to model long-term temporal dependencies, yielding strong performance. However, they tend to overlook the sequential nature of time series, which is crucial for maintaining temporal consistency. MLP-based methods (Hu et al., 2024; Yu et al., 2024), with their dense weights, capture interactions between different time steps in a simple but effective way. Their straightforward architecture makes them a popular choice for time series forecasting tasks. Additionally, GNN-based methods (Wu et al., 2020; Huang et al., 2023) are applied to capture the relationships between different variables in a graph structure.

## 2.2. Dependency Modeling

Regarding modeling dependencies in MTSF, some models adopt a Channel Independence (CI) strategy (Nie et al., 2023; Zeng et al., 2023; Lin et al., 2024), as in Figure 2(a), preserving only the temporal dependencies. These models use only the historical information of each individual sequence, ignoring the interactions between variables. Although the CI approach is robust, it wastes the potentially relevant relationships from inter-channel information. In contrast, the Channel Dependency (CD) strategy (Liu et al., 2024b; Zhang & Yan, 2023; Yu et al., 2024), as in Figure 2(b), models the full spectrum of information, providing information gains but also including redundant dependencies, thus reducing robustness. Therefore, designing an adaptive dependency modeling strategy remains a significant challenge. Recent research (Chen et al., 2024; Qiu et al., 2024b) employs a Channel Clustering (CC) strategy to dynamically select modeling strategies based on channel similarity, as in Figure 2(d). For instance, variables $x_1$ and $x_3$ are chosen to form a CD cluster, where inter-channel relationships are modeled, while variable $x_2$ is assigned to a CI cluster, relying only on its own historical values. However, this coarse-grained approach fails to capture the dynamically evolving interactions. Therefore, as in Figure 2(c), we design a robust and fine-grained filtering mechanism to customize the dependencies needed for each time segment.

## 3. Preliminaries

### 3.1. Problem Definition

In the multivariate time series forecasting task, let $\mathbf{X} = \{\mathbf{x_1}, \mathbf{x_2}, ..., \mathbf{x_C}\} \in \mathbb{R}^{C \times L}$ be the input time series,

where $C$ denotes the number of channels and $L$ denotes the look-back horizon. $\mathbf{x_i} \in \mathbb{R}^L$ represents one of the channels. The objective is to construct a model that predicts the future sequences $\mathbf{Y} = \{\hat{\mathbf{y}}_1, \hat{\mathbf{y}}_2, ..., \hat{\mathbf{y}}_C\} \in \mathbb{R}^{C \times T}$, where $T$ denotes the forecasting horizon.

### 3.2. Dependency Matrix

The dependency matrix $\mathbf{M} \in \mathbb{R}^{n \times n}$ describes the dependencies within channels at different time patches as the adjacency matrix of a spatial-temporal graph $\mathcal{G} = \{\mathcal{V}, \mathcal{E}\}$ with $\mathcal{V}$ as the node set and $\mathcal{E}$ as the edge set, where $n = C \times N$ and $N$ denotes the number of patches in the look-back sequence. $\mathbf{M}$ is utilized to depict the inter-node relationship, where $\mathbf{M}[i, j]$ indicates the weight of edge $e_{ij} \in \mathcal{E}$.

Moreover, we decompose $\mathcal{G}$ into patch-specific ego graphs $\{\mathcal{G}_i\}_1^n$. Each central node of the graph $\mathcal{G}_i$ represents the specific patch of input sequence. Then each ego graph $\mathcal{G}_i$ is divided into three subgraphs: spatial subgraph $\mathcal{G}_i^S$, temporal subgraph $\mathcal{G}_i^T$, and spatial-temporal subgraph $\mathcal{G}_i^{ST}$. Each subgraph describes a different type of dependency of a specific patch. The adjacency matrix of each subgraph can be obtained by applying masks to $\mathbf{M}$.

## 4. Method

The core of TimeFilter, as illustrated in Figure 3, is to capture fine-grained spatial-temporal dependencies through three key components: **(i) Spatial-Temporal Construction Module** segments the input sequence into non-overlapping patches and constructs the spatial-temporal graph; **(ii) Patch-Specific Filtration Module** filters out irrelevant information through the Mixture of Experts (MoE) with Dynamic Routing mechanism to obtain the spatial-temporal dependencies for the current sequence; **(iii) Adaptive Graph Learning Module** leverages the graph structure to aggregate pertinent information and derives the forecasting results. The details of each essential module are explained in the following subsections.

### 4.1. Spatial-Temporal Construction (STC) Module

In this stage, we construct the spatial-temporal graph using the input look-back sequence. First, each channel of the input sequence $\mathbf{X} \in \mathbb{R}^{C \times L}$ is divided into non-overlapping patches $\mathbf{X}'_p \in \mathbb{R}^{C \times N \times P}$, where each patch has a length of $P$ and the number of patches $N = \lceil \frac{L}{P} \rceil$. Then each patch is mapped to an embedded patch token $\mathbf{X}_p \in \mathbb{R}^{C \times N \times D}$.

$$\mathbf{X}'_p = \text{Patching}(X), \quad \mathbf{X}_p = \text{Embedding}(\mathbf{X}'_p) \quad (1)$$

The Embedding$(\cdot)$ operation transforms each patch from its original length $P$ to a hidden dimension $D$ through a trainable linear layer. In addition, we flatten the first two dimensions to obtain a set of spatial-temporal patches with

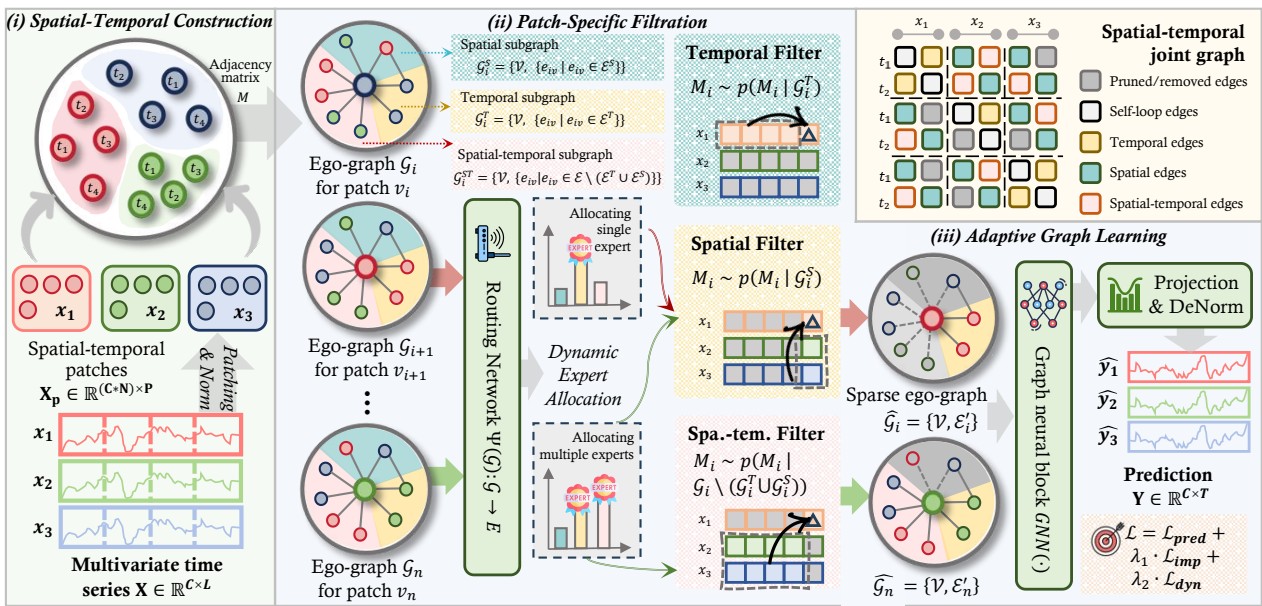

*Figure 3.* The overall structure of TimeFilter, which consists of: (i) **Spatial-Temporal Construction** is devised to construct the spatial-temporal graph from the input $\mathbf{X}$; (ii) **Patch-Specific Filtration** facilitates spatial-temporal dependencies by filtering out irrelevant information for each patch; (iii) **Adaptive Graph Learning** is leveraged to predict the future $\mathbf{Y}$ based on GNN.

$n = C \times N$ patches containing the enhanced local information $\mathbf{X}_p \in \mathbb{R}^{n \times D}$.

Next, we calculate the projection distances in a multi-head manner, allowing for richer similarity representation. We first divide $\mathbf{X}_p$ into different heads $\mathbf{X}_h \in \mathbb{R}^{H \times n \times \lfloor \frac{D}{H} \rfloor}$ and then calculate the projection distances $\text{Dist}(\cdot)$ by dot product. Here, $H$ is the number of heads and $A^T$ is the transpose.

$$\text{Dist}(\mathbf{X}_h) = \text{Linear}(\mathbf{X}_h) * \text{Linear}(\mathbf{X}_h)^T \in \mathbb{R}^{H \times n \times n} \quad (2)$$

Subsequently, based on above distances, we construct the graph with the $k$-Nearest Neighbor ($k$-NN) method. Each spatial-temporal patch is considered as a node, resulting in a total of $n$ nodes. For each node, we retain the $k$ nearest neighbors. In practice, we retain the $k = \lfloor \alpha * n \rfloor$ nearest neighbors, where $\alpha$ is a hand-tuned scaling factor. Formally, the adjacency matrix $\mathbf{M}$ can formulated as follows

$$\mathbf{M} = k\text{-NN}(\text{GeLU}(\text{Dist}(\mathbf{X}_h)), \alpha) \quad (3)$$

Instead of using the whole graph $\mathcal{G}$ with non-negligible noise (Huang et al., 2023), we decompose $\mathcal{G}$ into $n$ patch-specific ego graphs $\{\mathcal{G}_i\}_1^n$ to effectively resist noise, each corresponding to a patch. Each ego graph is further divided into three regions: temporal, spatial and spatial-temporal.

$$\mathcal{G}_i^T = \{\mathcal{V}, \{e_{iv}|e_{iv} \in \mathcal{E}_i^T\}\} \quad (4)$$

$$\mathcal{G}_i^S = \{\mathcal{V}, \{e_{iv}|e_{iv} \in \mathcal{E}_i^S\}\} \quad (5)$$

$$\mathcal{G}_i^{ST} = \{\mathcal{V}, \{e_{iv}|e_{iv} \in \mathcal{E} \setminus (\mathcal{E}^S \cup \mathcal{E}_i^T)\}\} \quad (6)$$

Here $\mathcal{E}_i^S$ is the spatial edge set, $\mathcal{E}_i^T$ is the temporal edge set, $e_v$ is one of the edges and $\forall i \in \{1, 2, ..., n\}$. The adjacency matrix of each subgraph $\mathbf{M}_i$ can be obtained by applying different masks to $\mathbf{M}$.

With the above designs, we obtain $n$ ego graphs $\mathcal{G}_i$ and the corresponding adjacency matrix $\mathbf{M}_i$ to represent the correlations of each patch.

### 4.2. Patch-Specific Filtration (PSF) Module

After constructing the graph, each edge connects two patch nodes, representing their dependency relationship. At this stage, we design a patch-specific MoE module composed of *routing network*, *dynamic expert allocation* and *filtration* to identify effective dependencies, filter out irrelevant ones, and obtain the spatial-temporal relationship.

**Routing Network.** We observe that time series data from different domains exhibit distinct characteristics and underlying structures, necessitating the modeling of different types of dependencies. Therefore, for each ego graph $\mathcal{G}_i$, we design three experts for different dependencies, including Temporal Filter, Spatial Filter, and Spa.-tem. Filter. Based on the router's decisions, filters with different dependency regions are assigned to different patches. Specifically, the filtration operation can be formulated as follows:

- **Temporal Filter** $\mathbf{M}_i \sim p(\mathbf{M}_i|\mathcal{G}_i^T)$ if the selection includes a Temporal Filter, the temporal edges $\mathcal{E}^T$ are re-

tained. For example, user behavior data with low inter-channel correlation benefits more from temporal dependencies (Yuan et al., 2022), which leverage the historical information of individual users.

- **Spatial Filter** $\mathbf{M}_i \sim p(\mathbf{M}_i|\mathcal{G}_i^S)$ If the selection includes a Spatial Filter, the spatial edges $\mathcal{E}^S$ are retained. This can be crucial for scenarios where spatial dependencies reflect the synchronization between multiple variables in real-time monitoring.

- **Spa.-tem. Filter**: $\mathbf{M}_i \sim p(\mathbf{M}_i|\mathcal{G}_i^{ST})$ If the selection includes a Spatial-temporal Filter, the spatial edges $\mathcal{E}\backslash(\mathcal{E}^S \cup \mathcal{E}_i^T)$ are retained. For instance, in traffic flow prediction, spatial-temporal dependencies are pivotal (Wang et al., 2024b), capturing the nuanced interactions between traffic dynamics across different locations and their evolution over time.

Following the classic concept of a (sparsely-gated) mixture-of-experts (Shazeer et al., 2017b; Zhang et al., 2024a) and graph sparsification paradigms (Wang et al., 2024a; Zhang et al., 2024b;c), we calculate the confidence $r$ of the current patch for each filter using a noisy gating mechanism.

$$r(\mathcal{G}_i) = \text{Softmax}(\psi(\mathbf{M}_i)), \tag{7}$$

$$\psi(\mathbf{M}_i) = \text{Linear}_g(\mathbf{M}_i) + \epsilon \cdot \text{Softplus}(\text{Linear}_n(\mathbf{M}_i)) \tag{8}$$

where $\psi(\mathbf{M}) \in \mathbb{R}^m$ is the calculated scores of patches for all filters, $\epsilon \in \mathcal{N}(0,1)$ is denotes the standard Gaussian noise, $\text{Linear}_g$ and $\text{Linear}_n$ are clean and noisy scores mapping, respectively. $m$ is the number of filters and in TimeFilter $m = 3$.

**Dynamic Expert Allocation.** The classical Top-$K$ routing MoE architecture assumes that the same number of experts is assigned to each input token, but this overlooks the differences between inputs. Specifically, in dependency modeling, certain time patches require a different number of dependencies. For example, when there is a clear lead-lag effect in stock data, both temporal and spatial-temporal dependencies are needed, while during unexpected events, spatial dependencies are more crucial to capture how stocks from the same sector react.

To adaptively customize the dependency relationships for each patch, we propose Dynamic Expert Allocation module to select filters based on the confidence of the patch itself, following MoE-Dynamic (Huang et al., 2024). Unlike the Top-$K$ routing mechanism, which selects a fixed number of experts, our method allows the model to evaluate whether the current selection of dependencies is sufficient. If not, it will continue to allocate more dependencies.

Thus, we rank the confidence $r$ from largest to smallest, resulting in a sorted index list $I$. Afterward, we find the smallest set of filters whose cumulative probability exceeds the threshold Top-$p$, and the number of selected filters $q$ is calculated by:

$$q(\mathcal{G}_i) = \underset{j \in \{1,2,...,m\}}{\arg\min} \sum_{k \le j} r(\mathcal{G}_i)_k \ge \text{Top-}p \tag{9}$$

Next, the calculation of output routing is:

$$g(\mathcal{G}_i)_j = \begin{cases} r(\mathcal{G}_i)_j, & \text{Filter}_j \in S_i \\ 0, & \text{Filter}_j \notin S_i \end{cases}, j \in \{1, 2, ..., m\} \tag{10}$$

where $S$ is the set of selected filters controlled by $q(\mathcal{G}_i)$ in Eq. 9:

$$S_i = \{\text{Filter}_{I_1}, \text{Filter}_{I_2}, ..., \text{Filter}_{I_{q(\mathcal{G}_i)}}\} \tag{11}$$

**Filtration.** Finally, each ego graph $\mathcal{G}_i$, $\forall i \in \{1, 2, ..., n\}$, based on output dynamic routing $g(\mathcal{G}_i) \in \mathbb{R}^m$, we perform a filtration operation on $\mathbf{M}_i$ to obtain the adjacency matrix $\mathbf{M}_i^{'}$ and the corresponding edge set $\mathcal{E}_i^{'}$ and the sparse graph $\widehat{\mathcal{G}}_i$ after filtering out redundant dependencies. For example, if $S_i = \{\text{Temporal Filter}, \text{Spa.-tem. Filter}\}$, the ego graph after filtering will be:

$$\widehat{\mathcal{G}}_i = \{\mathcal{V}, \{e_{iv}|e_{iv} \in \mathcal{E}_i^T \cup \mathcal{E}\backslash(\mathcal{E}^S \cup \mathcal{E}_i^T)\}\} \tag{12}$$

It is worth noting that the filtration operation on the $n$ ego-graphs can be performed in parallel without introducing additional computational complexity.

### 4.3. Adaptive Graph Learning (AGL) Module

After obtaining $\widehat{\mathcal{G}}_i = \{\mathcal{V}, \mathcal{E}_i^{'}\}$, we restore the $n$ ego graphs into a global graph $\widehat{\mathcal{G}}$ and aggregate the $\mathbf{M}_i^{'}$ into corresponding $\mathbf{M}^{'}$.

$$\widehat{\mathcal{G}} = \bigcup_1^n \widehat{\mathcal{G}}_i = \{\mathcal{V}, \bigcup_1^n \mathcal{N}(v_i)\} \tag{13}$$

where $\mathcal{N}(\cdot)$ is the set of neighboring nodes of a given node.

Then, the graph learning layer learns the graph adjacency matrix $\mathbf{M}^{'}$ adaptively to capture the hidden relationships among time series data.

$$h_i^{(l)} = \text{COMB}(h_i^{(l-1)}, \text{AGGR}\{h_j^{(k-1)} : v_j \in \mathcal{N}(v_i)\}) \tag{14}$$

where $0 \le l \le O$, $O$ is the number of GNN layers, $h_i^{(0)} = \mathbf{X}_h \in \mathbb{R}^{H \times n \times \lfloor \frac{D}{H} \rfloor}$ and $h_i^{(l)}(1 \le l \le O)$ is the embedding of node $v_i$ at the $l$-th layer. $\text{AGGR}(\cdot)$ and $\text{COMB}(\cdot)$ are the functions used for aggregating neighborhood information and combining ego- and neighbor-representations, respectively. The $\text{AGGR}(\cdot)$ employed in TimeFilter is an additive operation, where the representation of each node is updated by summing the features of its neighboring nodes.

Finally, to perform forecasting, our model utilizes a residual module and a linear projection in time dimension. This transformation can be expressed as:

$$\mathbf{Y} = \text{Linear}(\mathbf{X}_h + h^{(O)}) \in \mathbb{R}^{C \times T} \tag{15}$$

## 4.4. Loss Function

The loss function of TimeFilter consists of three components. For the predictors, the Mean Squared Error (MSE) loss ($\mathcal{L}_{\text{pred}} = \frac{1}{T} \sum_{i=0}^{T} \|\mathbf{x}_{:,\mathbf{i}} - \hat{\mathbf{x}}_{:,\mathbf{i}}\|_2^2$) is used to measure the variance between predicted values and ground truth. Moreover, following dynamic routing (Huang et al., 2024), we introduce an dynamic loss $\mathcal{L}_{\text{dyn}}$ to avoid assigning low confidence to all experts, thereby activating a larger number of experts to achieve better performance.

$$\mathcal{L}_{\text{dyn}} = -\frac{1}{n} \sum_{i=1}^{n} \sum_{j=1}^{m} r(\mathcal{G}_i)_j * \log(r(\mathcal{G}_i)_j) \qquad (16)$$

Moreover, we introduce an expert importance loss $\mathcal{L}_{\text{imp}}$ (Shazeer et al., 2017a) to prevent TimeFilter from converging to a trivial solution where only a single group of experts is consistently selected:

$$\mathcal{L}_{\text{imp}} = \frac{1}{n} \sum_{i=1}^{n} \frac{\text{Std}(r(\mathcal{G}_i))}{\text{Mean}(r(\mathcal{G}_i)) + \epsilon} \qquad (17)$$

where $\epsilon$ is a small positive constant to prevent numerical instability, $\text{Std}(\cdot)$ and $\text{Mean}(\cdot)$ calculate the standard deviation and the mean, respectively. Therefore, the final loss function is defined as:

$$\mathcal{L} = \mathcal{L}_{\text{pred}} + \lambda_1 \mathcal{L}_{\text{dyn}} + \lambda_2 \mathcal{L}_{\text{imp}}, \qquad (18)$$

where $\lambda_1$ and $\lambda_2$ are the scaling factors.

## 5. Experiments

### 5.1. Experimental Setup

**Datasets.** For long-term forecasting, we conduct extensive experiments on selected 9 commonly real-world datasets from various application scenarios, including ETT (ETTh1, ETTh2, ETTm1, ETTm2), Traffic, Electricity, Weather, Solar-Energy (Miao et al., 2024a;b) and Climate (with 1763 channels) datasets. For short-term forecasting, we adopt 4 benchmarks from PEMS (PEMS03, PEMS04, PEMS07, PEMS08) (Liu et al., 2025b;a). The statistics of the dataset are shown in Appendix A.1.

**Baselines.** We compare our method with SOTA representative methods, including GNN-based methods: MSGNet (Cai et al., 2024), CrossGNN (Huang et al., 2023); Transformer-based methods: CATS (Kim et al., 2024), iTransformer (Liu et al., 2024b), PatchTST (Nie et al., 2023), Crossformer (Zhang & Yan, 2023), FEDformer (Zhou et al., 2022); Linear-based methods: Leddam (Yu et al., 2024), SOFTS (Han et al., 2024a), DUET (Qiu et al., 2024b), DLinear (Zeng et al., 2023), TiDE (Das et al., 2023); CNN-based methods: ModernTCN (Donghao & Xue, 2024), TimesNet (Wu et al., 2023), SCINet (Liu et al., 2022). In addition,

CCM (Chen et al., 2024) is included as a plug-in in the baseline as PatchTST+CCM, or CCM for short.

**Implementation Details.** All experiments are implemented in PyTorch (Paszke et al., 2019) and conducted on one NVIDIA A100 40GB GPU. We use the Adam optimizer (Kingma, 2014) with a learning rate selected from $\{1e^{-3}, 1e^{-4}, 5e^{-4}\}$. The Top-$p$ in dynamic expert allocation is selected from $\{0.0, 0.5\}$. We select two common metrics in time series forecasting: Mean Absolute Error (MAE) and Mean Squared Error (MSE). For additional details on hyperparameters and settings, please refer to Appendix A.3.

### 5.2. Results

**Long-term Forecasting.** We report the results of the long-term forecasting in Table 1. The input length $L$ is 96 for our method and all baselines. The forecasting horizon $T$ is $\{96, 192, 336, 720\}$. The results show that TimeFilter adaptively models various dependencies for each dataset in different domains, and has been proven to yield significant improvements. Compared to the second-best model Leddam (Yu et al., 2024), TimeFilter reduces MSE and MAE by $4.48\%/2.23\%$, respectively.

Additionally, we perform the Wilcoxon test with Leddam (Yu et al., 2024) and obtain the p-value of $4.66e^{-10}$, indicating a significant improvement at the $99\%$ confidence level. The detailed error bars are shown in Appendix B.1.

Furthermore, according to the scaling law (Shi et al., 2024) of TSF, for a given dataset size, as the look-back horizon $L$ increases, so does the noise in the input sequence, which does not necessarily improve the performance of the model. In addition, the dataset size has more influence on the optimal horizon, while the model size has less influence. Therefore, we conduct a search for different look-back horizons to determine the optimal horizon for the 4 relatively large datasets and compare TimeFilter with other baselines. As shown in Table 2, TimeFilter demonstrates robust resistance to noise and maintains SOTA performance even with longer optimal horizon inputs, outperforming other methods. Specifically, compared to the channel clustering strategy methods DUET (Qiu et al., 2024b) and CCM (Chen et al., 2024), TimeFilter reduces MSE and MAE by $5.34\%/1.40\%$ and $6.89\%/3.69\%$, respectively. This demonstrates the effectiveness of our fine-grained adaptive filtration strategy.

**Short-term Forecasting.** We report the results of the short-term forecasting in Table 3. The input length $L$ is 96 for our method and all baselines. The forecasting horizon $T$ is $\{12, 24, 48\}$. From the table, TimeFilter consistently outperforms other methods across all 4 PEMS datasets. Specifically, on PEMS08, TimeFilter reduces MSE and MAE by $13.54\%/3.13\%$ compared to the second-best DUET (Qiu

| Models | TimeFilter (Ours) | | DUET CC (2024b) | | iTransformer CD (2024b) | | Leddam CD (2024) | | MSGNet CD (2024) | | SOFTS (2024a) | | CrossGNN CD (2023) | | PatchTST CI (2023) | | Crossformer CD (2023) | | TimesNet CD (2023) | | DLinear CI (2023) | | FEDformer CD (2022) | |
|---|---|---|---|---|---|---|---|---|---|---|---|---|---|---|---|---|---|---|---|---|---|---|---|---|
| Metric | MSE | MAE | MSE | MAE | MSE | MAE | MSE | MAE | MSE | MAE | MSE | MAE | MSE | MAE | MSE | MAE | MSE | MAE | MSE | MAE | MSE | MAE | MSE | MAE |
| ETT (Avg.) | **0.358** | **0.385** | 0.371 | 0.388 | 0.383 | 0.399 | 0.368 | 0.388 | 0.384 | 0.403 | 0.379 | 0.396 | 0.371 | 0.392 | 0.381 | 0.397 | 0.685 | 0.578 | 0.391 | 0.404 | 0.446 | 0.447 | 0.408 | 0.428 |
| Weather | **0.239** | **0.269** | 0.251 | 0.273 | 0.258 | 0.279 | 0.242 | 0.272 | 0.249 | 0.278 | 0.255 | 0.278 | 0.247 | 0.289 | 0.259 | 0.281 | 0.259 | 0.315 | 0.259 | 0.287 | 0.265 | 0.315 | 0.309 | 0.360 |
| Electricity | **0.158** | **0.256** | 0.172 | 0.258 | 0.178 | 0.270 | 0.169 | 0.263 | 0.194 | 0.300 | 0.174 | 0.264 | 0.201 | 0.300 | 0.216 | 0.304 | 0.244 | 0.334 | 0.193 | 0.295 | 0.225 | 0.319 | 0.214 | 0.327 |
| Traffic | **0.407** | 0.268 | 0.451 | 0.269 | 0.428 | 0.282 | 0.467 | 0.294 | 0.641 | 0.370 | 0.409 | **0.267** | 0.583 | 0.323 | 0.555 | 0.362 | 0.550 | 0.304 | 0.620 | 0.336 | 0.625 | 0.383 | 0.610 | 0.376 |
| Solar-Energy | **0.223** | 0.250 | 0.237 | **0.233** | 0.233 | 0.262 | 0.230 | 0.264 | 0.262 | 0.288 | 0.229 | 0.256 | 0.324 | 0.377 | 0.270 | 0.307 | 0.641 | 0.639 | 0.301 | 0.319 | 0.330 | 0.401 | 0.292 | 0.381 |

*Table 1.* Long-term forecasting results. The input length $L$ is 96. All results are averaged across four different forecasting horizons: $T \in \{96, 192, 336, 720\}$. The best and second-best results are in **bold** and underlined, respectively. See Table 8 for full results.

| Models | TimeFilter (Ours) | | DUET CC (2024b) | | CCM CC (2024) | | PDF CI (2024) | | ModernTCN CD (2024) | | iTransformer CD (2024b) | | CATS CD (2024) | | PatchTST CI (2023) | | Crossformer CD (2023) | | TimesNet CD (2023) | | DLinear CI (2023) | |
|---|---|---|---|---|---|---|---|---|---|---|---|---|---|---|---|---|---|---|---|---|---|---|
| Metric | MSE | MAE | MSE | MAE | MSE | MAE | MSE | MAE | MSE | MAE | MSE | MAE | MSE | MAE | MSE | MAE | MSE | MAE | MSE | MAE | MSE | MAE |
| Weather | **0.216** | 0.258 | 0.218 | **0.252** | 0.225 | 0.263 | 0.225 | 0.261 | 0.224 | 0.264 | 0.232 | 0.270 | 0.244 | 0.272 | 0.226 | 0.264 | 0.234 | 0.292 | 0.255 | 0.282 | 0.242 | 0.293 |
| Electricity | **0.150** | **0.246** | 0.158 | 0.248 | 0.167 | 0.261 | 0.156 | 0.250 | 0.156 | 0.253 | 0.163 | 0.258 | 0.178 | 0.264 | 0.159 | 0.253 | 0.181 | 0.279 | 0.192 | 0.295 | 0.166 | 0.264 |
| Traffic | **0.360** | **0.254** | 0.394 | 0.257 | 0.389 | 0.259 | 0.377 | 0.256 | 0.396 | 0.270 | 0.397 | 0.281 | 0.449 | 0.283 | 0.391 | 0.264 | 0.523 | 0.284 | 0.620 | 0.336 | 0.434 | 0.295 |
| Climate | **1.047** | **0.490** | 1.103 | 0.509 | 1.123 | 0.512 | 1.315 | 0.555 | 1.155 | 0.542 | 1.060 | 0.496 | 1.095 | 0.509 | 1.387 | 0.562 | 1.170 | 0.550 | 1.295 | 0.569 | 1.134 | 0.541 |

*Table 2.* Long-term forecasting results. The input length $L$ is searched from $\{192, 336, 512, 720\}$ for optimal horizon in the scaling law of TSF (Shi et al., 2024). All results are averaged across four different forecasting horizons: $T \in \{96, 192, 336, 720\}$. The best and second-best results are in **bold** and underlined, respectively. See Table 9 for full results.

| Models | TimeFilter (Ours) | | DUET CC (2024b) | | iTransformer CD (2024b) | | Leddam CD (2024) | | SOFTS (2024a) | | PatchTST CI (2023) | | Crossformer CD (2023) | | TimesNet CD (2023) | | TiDE CD (2023) | | DLinear CI (2023) | | SCINet CD (2022) | | FEDformer CD (2022) | |
|---|---|---|---|---|---|---|---|---|---|---|---|---|---|---|---|---|---|---|---|---|---|---|---|---|
| Metric | MSE | MAE | MSE | MAE | MSE | MAE | MSE | MAE | MSE | MAE | MSE | MAE | MSE | MAE | MSE | MAE | MSE | MAE | MSE | MAE | MSE | MAE | MSE | MAE |
| PEMS03 | **0.084** | **0.191** | 0.086 | 0.192 | 0.096 | 0.204 | 0.101 | 0.210 | 0.087 | 0.192 | 0.151 | 0.265 | 0.138 | 0.253 | 0.119 | 0.271 | 0.271 | 0.380 | 0.219 | 0.295 | 0.093 | 0.203 | 0.167 | 0.291 |
| PEMS04 | **0.083** | **0.186** | 0.096 | 0.203 | 0.098 | 0.207 | 0.102 | 0.213 | 0.091 | 0.196 | 0.162 | 0.273 | 0.145 | 0.267 | 0.109 | 0.220 | 0.307 | 0.405 | 0.236 | 0.350 | 0.085 | 0.194 | 0.195 | 0.308 |
| PEMS07 | **0.071** | **0.170** | 0.076 | 0.176 | 0.088 | 0.190 | 0.087 | 0.192 | 0.075 | 0.173 | 0.166 | 0.270 | 0.181 | 0.272 | 0.106 | 0.208 | 0.297 | 0.394 | 0.241 | 0.343 | 0.112 | 0.211 | 0.133 | 0.282 |
| PEMS08 | **0.083** | **0.186** | 0.096 | 0.192 | 0.127 | 0.212 | 0.102 | 0.211 | 0.114 | 0.208 | 0.238 | 0.289 | 0.232 | 0.270 | 0.150 | 0.244 | 0.347 | 0.421 | 0.281 | 0.366 | 0.133 | 0.225 | 0.234 | 0.326 |

*Table 3.* Short-term forecasting results. The input length $L$ is 96. All results are averaged across three different forecasting horizons: $T \in \{12, 24, 48\}$. The best and second-best results are in **bold** and underlined, respectively. See Table 10 for full results.

et al., 2024b). On PEMS07 with 883 channels, TimeFilter reduces MSE and MAE by 5.33%/1.73% compared to the second-best SOFTS (Han et al., 2024a).

### 5.3. Ablation Study

To validate the effectiveness of TimeFilter, we conduct a comprehensive ablation study on its architectural design.

**Comparison of different Filter methods.** We carefully design six other filtering methods and the comparison is shown in the Filter Replace section of Table 4. **Top-$K$** strategy selects the top $K$ edges with the largest weights from the ego graph $\mathcal{G}_i$ of each patch, while **Random-$K$** randomly selects $K$ edges. **RegionTop-$K$** strategy selects the top $K$ edges with the largest weights within the region corresponding to three types of dependencies, while **RegionThre** learns a threshold for each region through a linear mapping and selects edges with weights greater than the threshold in each region. **C-Filter** refers to channel-wise filtering, where

all patches within the look-back sequence are filtered based on the same type of dependency. The **w/o Filter** refers to the case where no filtering is applied after graph construction.

The result shows that TimeFilter consistently outperforms other filtering methods. We conclude that higher-weight relationships may be due to spurious regression (Liu et al., 2024a) rather than effective dependencies. Furthermore, time series data from different domains require different types of relationships.

**MoE component ablation.** We conduct an ablation study on the routing mechanism (Static Routing with only the most significant filter selected) and $\mathcal{L}_{\text{imp \& dyn}}$, demonstrating the importance of dynamic selection and load balancing.

### 5.4. Model Analysis

**Case study on dependency modeling.** As shown in Figure 4, we visualize the dependencies and distribution of

| Catagories | | | | | | | Filter Replace | | | | | | | | | MoE | | |
|---|---|---|---|---|---|---|---|---|---|---|---|---|---|---|---|---|---|---|
| Cases | TimeFilter | | Top-$K$ | | Random-$K$ | | RegionTop-$K$ | | RegionThre | | C-Filter | | w/o Filter | | Static Routing | | w/o $\mathcal{L}_{\text{imp \& dyn}}$ | |
| | MSE | MAE | MSE | MAE | MSE | MAE | MSE | MAE | MSE | MAE | MSE | MAE | MSE | MAE | MSE | MAE | MSE | MAE |
| Weahter | **0.239** | **0.269** | 0.243 | 0.272 | 0.243 | 0.272 | 0.241 | 0.271 | 0.241 | 0.270 | 0.242 | 0.271 | 0.244 | 0.273 | 0.242 | 0.271 | 0.243 | 0.272 |
| Electricity | **0.158** | **0.256** | 0.167 | 0.263 | 0.168 | 0.264 | 0.168 | 0.263 | 0.172 | 0.267 | 0.170 | 0.265 | 0.167 | 0.263 | 0.164 | 0.260 | 0.163 | 0.259 |
| Traffic | **0.407** | **0.268** | 0.413 | 0.271 | 0.412 | 0.271 | 0.411 | 0.270 | 0.412 | 0.271 | 0.413 | 0.271 | 0.414 | 0.273 | 0.410 | 0.271 | 0.410 | 0.271 |
| Solar | **0.223** | **0.250** | 0.231 | 0.258 | 0.231 | 0.258 | 0.233 | 0.257 | 0.235 | 0.259 | 0.234 | 0.258 | 0.234 | 0.262 | 0.228 | 0.258 | 0.227 | 0.258 |
| ETTh1 | **0.426** | **0.431** | 0.441 | 0.438 | 0.441 | 0.437 | 0.441 | 0.438 | 0.437 | 0.436 | 0.431 | 0.436 | 0.442 | 0.438 | 0.440 | 0.434 | 0.430 | 0.432 |
| ETTm2 | **0.272** | **0.321** | 0.276 | 0.324 | 0.276 | 0.323 | 0.273 | 0.323 | 0.276 | 0.324 | 0.277 | 0.325 | 0.277 | 0.324 | 0.274 | 0.322 | 0.273 | 0.322 |
| PEMS08 | **0.083** | **0.186** | 0.086 | 0.191 | 0.085 | 0.189 | 0.085 | 0.190 | 0.085 | 0.189 | 0.086 | 0.190 | 0.086 | 0.191 | 0.084 | 0.188 | 0.084 | 0.187 |

*Table 4.* Ablation study of TimeFilter. The The input length $L$ is 96. See Table 11 for full results.

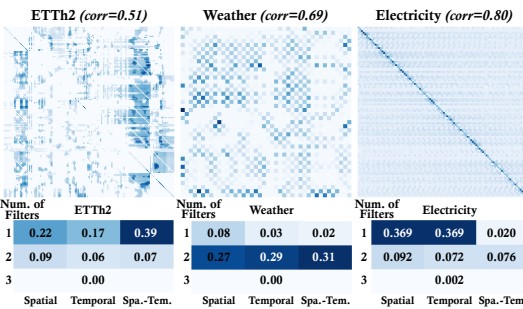

*Figure 4.* Above are the dependencies learned by TimeFilter from the ETTh2, Weather and Electricity datasets. Below is the distribution of selected filters, where the x-axis represents the dependency types and the y-axis represents the number of different filters.

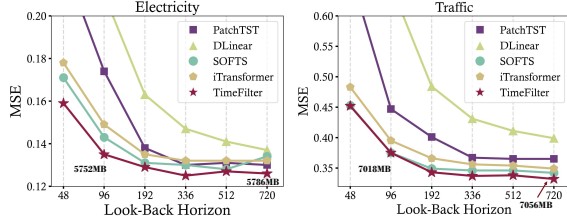

*Figure 5.* Influence of look-back horizon. TimeFilter consistently outperforms other models under different look-back horizons.

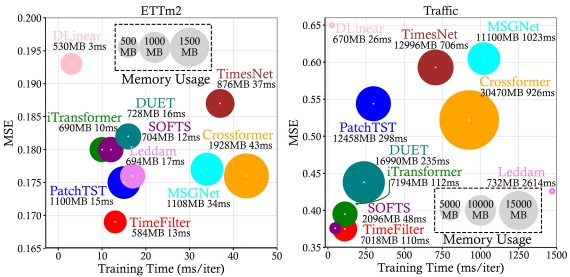

*Figure 6.* Model efficiency comparison under input-96-predict-96 of ETTm2 and Traffic datasets.

selected filters of one batch in the ETTh2, Weather, and Electricity datasets. Indeed, it shows that different datasets require different types of dependency and that TimeFilter can capture these dependencies powerfully. Furthermore, as the correlation (Qiu et al., 2024a) within datasets increases, TimeFilter tends to utilize more effective dependencies to capture future variations. TimeFilter customizes the filtering of redundant dependencies for each dataset, enhancing its ability to express omni-series representations.

**Influence of look-back horizon.** To find the optimal horizon (Shi et al., 2024), we explore how varying the length of look-back horizons from $\{48, 96, 192, 336, 512, 720\}$ impacts the forecasting performance. As shown in Figure 5, TimeFilter effectively resists noise and exploits vital information in longer sequences, outperforming other models over different look-back horizons. Furthermore, as the hyperparameter length of the patch $P$ increases proportionally with the look-back horizon $L$, the memory usage and inference efficiency of TimeFilter remain almost unchanged, thus preventing memory explosion in extremely long sequences.

**Efficiency Analysis.** We comprehensively compare the forecasting performance, training speed, and memory footprint of TimeFilter and other baselines, using the official

model configurations and the same batch size. As shown in Figure 6, the efficiency of TimeFilter exceeds other Transformer-based, CNN-based and GNN-based methods.

# 6. Conclusion

Considering the complex dynamic interactions inherent in real-world multivariate time series, we propose TimeFilter to adaptively, finely, and robustly model the dependencies. Technically, we design patch-specific filtration to effectively remove redundant relationships, thereby exploiting the most important dependencies for accurate forecasting. Extensive experiments show that TimeFilter consistently achieves SOTA performance in both long- and short-term forecasting tasks. Our work provides an in-depth exploration of dependency modeling in TSF, which we believe may pave the way for further research into dependency representations.

## Acknowledgements

Dai Tao is supported in part by the National Natural Science Foundation of China, under Grant (62302309, 62171248), Shenzhen Science and Technology Program (JCYJ20220818101014030, JCYJ20220818101012025).

## Impact Statement

Time series forecasting plays a crucial role in a variety of real-world domains, such as finance, weather forecasting, and traffic control. Our study presents a novel method for enhancing the dependency modeling. The datasets utilized are publicly accessible, promoting transparency and reproducibility. We do not anticipate any ethical issues arising from this work. The primary contributions of this research advance the field of time series forecasting, with potential applications in multiple sectors.

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

# A. Implementation Details

## A.1. Datasets

We conduct extensive experiments on eight widely-used time series datasets for long-term forecasting and PEMS datasets for short-term forecasting. We report the statistics in Table 5. Detailed descriptions of these datasets are as follows:

(1) **ETT** (Electricity Transformer Temperature) dataset (Zhou et al., 2021) encompasses temperature and power load data from electricity transformers in two regions of China, spanning from 2016 to 2018. This dataset has two granularity levels: ETTh (hourly) and ETTm (15 minutes).

(2) **Weather** dataset (Wu et al., 2023) captures 21 distinct meteorological indicators in Germany, meticulously recorded at 10-minute intervals throughout 2020. Key indicators in this dataset include air temperature, visibility, among others, offering a comprehensive view of the weather dynamics.

(3) **Electricity** dataset (Wu et al., 2023) features hourly electricity consumption records in kilowatt-hours (kWh) for 321 clients. Sourced from the UCL Machine Learning Repository, this dataset covers the period from 2012 to 2014, providing valuable insights into consumer electricity usage patterns.

(4) **Traffic** dataset (Wu et al., 2023) includes data on hourly road occupancy rates, gathered by 862 detectors across the freeways of the San Francisco Bay area. This dataset, covering the years 2015 to 2016, offers a detailed snapshot of traffic flow and congestion.

(5) **Solar-Energy** dataset (Lai et al., 2018) contains solar power production data recorded every 10 minutes throughout 2006 from 137 photovoltaic (PV) plants in Alabama.

(6) **Climate** dataset (Godahewa et al., 2021) records regional precipitation data across 1,763 stations in Australia.

(7) **PEMS** dataset (Liu et al., 2022) comprises four public traffic network datasets (PEMS03, PEMS04, PEMS07, and PEMS08), constructed from the Caltrans Performance Measurement System (PeMS) across four districts in California. The data is aggregated into 5-minute intervals, resulting in 12 data points per hour and 288 data points per day.

| Tasks | Dataset | Dim | Prediction Length | Dataset Size | Frequency |
|---|---|---|---|---|---|
| | ETTm1 | 7 | $\{96, 192, 336, 720\}$ | $(34465, 11521, 11521)$ | 15 min |
| | ETTm2 | 7 | $\{96, 192, 336, 720\}$ | $(34465, 11521, 11521)$ | 15 min |
| | ETTh1 | 7 | $\{96, 192, 336, 720\}$ | $(8545, 2881, 2881)$ | 15 min |
| Long-term | ETTh2 | 7 | $\{96, 192, 336, 720\}$ | $(8545, 2881, 2881)$ | 15 min |
| Forecasting | Weather | 21 | $\{96, 192, 336, 720\}$ | $(36792, 5271, 10540)$ | 10 min |
| | Electricity | 321 | $\{96, 192, 336, 720\}$ | $(18317, 2633, 5261)$ | 1 hour |
| | Traffic | 862 | $\{96, 192, 336, 720\}$ | $(12185, 1757, 3509)$ | 1 hour |
| | Solar-Energy | 137 | $\{96, 192, 336, 720\}$ | $(36601, 5161, 10417)$ | 10 min |
| | Climate | 1763 | $\{96, 192, 336, 720\}$ | $(6763, 989, 2070)$ | 10 min |
| | PEMS03 | 358 | 12 | $(15617, 5135, 5135)$ | 5 min |
| Short-term | PEMS04 | 307 | 12 | $(10172, 3375, 3375)$ | 5 min |
| Forecasting | PEMS07 | 883 | 12 | $(16911, 5622, 5622)$ | 5 min |
| | PEMS08 | 170 | 12 | $(10690, 3548, 265)$ | 5 min |

*Table 5.* Dataset detailed descriptions. "Dataset Size" denotes the total number of time points in (Train, Validation, Test) split respectively. "Prediction Length" denotes the future time points to be predicted. "Frequency" denotes the sampling interval of time points.

## A.2. Metric Details

Regarding metrics, we utilize the mean square error (MSE) and mean absolute error (MAE) for long-term forecasting. The calculations of these metrics are:

$$MSE = \frac{1}{T} \sum_{0}^{T} (\hat{y}_i - y_i)^2 \, , \qquad MAE = \frac{1}{T} \sum_{0}^{T} |\hat{y}_i - y_i|$$

## A.3. Experiment details

All experiments are implemented in PyTorch (Paszke et al., 2019) and conducted on an NVIDIA A100 40GB GPU. We use the Adam optimizer (Kingma, 2014). The batch size is set to 16 for the Electricity and Traffic datasets, and 32 for all other datasets. Table 6 provides detailed hyperparameter settings for each dataset.

| Tasks | Dataset | Length of Patch | e_layers | lr | d_model | d_ff | Num. of Epochs |
|---|---|---|---|---|---|---|---|
| Long-term Forecasting | ETTm1 | 8 | 2 | 1e-4 | 256 | 256 | 10 |
| | ETTm2 | 16 | 2 | 1e-4 | 128 | 128 | 10 |
| | ETTh1 | 2 | 2 | 1e-4 | 128 | 256 | 10 |
| | ETTh2 | 4 | 1 | 1e-4 | 128 | 256 | 10 |
| | Weather | 48 | 2 | 5e-4 | 128 | 256 | 10 |
| | Electricity | 32 | 2 | 1e-3 | 512 | 512 | 15 |
| | Traffic | 96 | 3 | 1e-3 | 512 | 2048 | 30 |
| | Solar-Energy | 48 | 2 | 5e-4 | 256 | 512 | 10 |
| Short-term Forecasting | PEMS03 | 48 | 2 | 1e-3 | 512 | 1024 | 20 |
| | PEMS04 | 48 | 2 | 5e-4 | 512 | 1024 | 20 |
| | PEMS07 | 96 | 2 | 1e-3 | 512 | 1024 | 20 |
| | PEMS08 | 48 | 2 | 1e-3 | 512 | 512 | 20 |

*Table 6.* Hyperparameter settings for different datasets. "e_layers" denotes the number of graph block. "lr" denotes the learning rate. "d_model" and "d_ff" denote the model dimension of attention layers and feed-forward layers, respectively.

# B. Full Results

## B.1. Error Bars

In this paper, we repeat all the experiments three times. Here we report the standard deviation of our proposed TimeFilter and the second best model Leddam (Yu et al., 2024), as well as the statistical significance test in Table 7. We perform the Wilcoxon test with Leddam (Yu et al., 2024) and obtain the p-value of $4.66e^{-10}$, indicating a significant improvement at the 99% confidence level.

## B.2. Long-term Forecasting

Table 8 and Table 9 present the full results for long-term forecasting, including both the results for the fixed look-back horizon $L = 96$ results and the results obtained through hyperparameter search for the optimal horizon according to the scaling law (Shi et al., 2024) of TSF. The hyperparameter search process involved exploring look-back horizons $L \in \{192, 336, 512, 720\}$. In both settings, TimeFilter consistently achieves the best performance, demonstrating its effectiveness and robustness. In particular, under the optimal horizon condition, TimeFilter can still adaptively model effective dependencies in long sequences.

## B.3. Short-term Forecasting

Table 10 present the full results for short-term forecasting. The look-back horizon $L$ is set to 96 and the forecasting horizon $T \in \{12, 24, 48\}$. Across all four PEMS datasets, TimeFilter consistently delivers the highest performance, particularly with a significant improvement on PEMS08, showcasing its effectiveness and reliability.

| Model | TimeFilter | | Leddam (2024) | | Confidence |
|---|---|---|---|---|---|
| Dataset | MSE | MAE | MSE | MAE | Interval |
| ETTm1 | $0.377 \pm 0.004$ | $0.393 \pm 0.006$ | $0.386 \pm 0.008$ | $0.397 \pm 0.006$ | 99% |
| ETTm2 | $0.272 \pm 0.004$ | $0.321 \pm 0.002$ | $0.281 \pm 0.003$ | $0.325 \pm 0.003$ | 99% |
| ETTh1 | $0.420 \pm 0.005$ | $0.428 \pm 0.003$ | $0.431 \pm 0.005$ | $0.429 \pm 0.004$ | 99% |
| ETTh2 | $0.364 \pm 0.003$ | $0.397 \pm 0.002$ | $0.373 \pm 0.009$ | $0.399 \pm 0.007$ | 99% |
| Weather | $0.239 \pm 0.006$ | $0.269 \pm 0.004$ | $0.242 \pm 0.009$ | $0.272 \pm 0.006$ | 99% |
| Electricity | $0.158 \pm 0.005$ | $0.256 \pm 0.006$ | $0.169 \pm 0.005$ | $0.263 \pm 0.005$ | 99% |
| Traffic | $0.407 \pm 0.008$ | $0.268 \pm 0.004$ | $0.467 \pm 0.006$ | $0.294 \pm 0.008$ | 99% |
| Solar | $0.223 \pm 0.006$ | $0.250 \pm 0.003$ | $0.230 \pm 0.008$ | $0.264 \pm 0.005$ | 99% |

*Table 7.* Standard deviation and statistical tests for TimeFilter and second-best method (Leddam) on ETT, Weather, Electricity, Traffic, and Solar datasets.

### B.4. Ablation Study

We present the full results of the ablation studies discussed in the main text in Table 11. We carefully design six other filtering methods to validate the effectiveness of TimeFilter.

(1) **Top-$K$** strategy selects the top $K$ edges with the largest weights from the ego graph $\mathcal{G}_i$ of each patch.

(2) **Random-$K$** randomly selects $K$ edges from the ego graph $\mathcal{G}_i$ of each patch.

(3) **RegionTop-$K$** strategy selects the top $K$ edges with the largest weights within the region corresponding to three types of dependencies.

(4) **RegionThre** learns a threshold for each region through a linear mapping and selects edges with weights greater than the threshold in each region.

(5) **C-Filter** refers to channel-wise filtering, where all patches within the look-back sequence are filtered based on the same type of dependency.

(6) **w/o Filter** refers to the case where no filtering is applied after graph construction.

## C. Visualization

In order to facilitate a clear comparison between different models, we present supplementary prediction examples for four representative datasets, as Electricity in Figure 7, Traffic in Figure 8, Weather in Figure 9 and PEMS08 in Figure 10, respectively. The look-back horizon $L$ is 96 for all four datasets. The forecasting horizon $T$ is 96 for Eleltricity, Traffic and Weather, while $T$ is 48 for PEMS08. Among the different models, TimeFilter delivers the most accurate predictions of future series variations and demonstrates superior performance.

| Models | | TimeFilter (Ours) | | DUET (2024b) | | iTransformer (2024b) | | Leddam (2024) | | MSGNet (2024) | | SOFTS (2024a) | | CrossGNN (2023) | | PatchTST (2023) | | Crossformer (2023) | | TimesNet (2023) | | DLinear (2023) | | FEDformer (2022) | |
|---|---|---|---|---|---|---|---|---|---|---|---|---|---|---|---|---|---|---|---|---|---|---|---|---|
| Metric | | MSE | MAE | MSE | MAE | MSE | MAE | MSE | MAE | MSE | MAE | MSE | MAE | MSE | MAE | MSE | MAE | MSE | MAE | MSE | MAE | MSE | MAE | MSE | MAE |
| ETTm1 | 96 | **0.313** | **0.354** | 0.324 | 0.354 | 0.334 | 0.368 | 0.319 | 0.359 | 0.319 | 0.366 | 0.325 | 0.361 | 0.335 | 0.373 | 0.329 | 0.367 | 0.404 | 0.426 | 0.338 | 0.375 | 0.346 | 0.374 | 0.379 | 0.419 |
| | 192 | **0.356** | 0.380 | 0.369 | 0.379 | 0.377 | 0.391 | 0.369 | 0.383 | 0.376 | 0.397 | 0.375 | 0.389 | 0.372 | 0.390 | 0.367 | 0.385 | 0.450 | 0.451 | 0.374 | 0.387 | 0.382 | 0.391 | 0.426 | 0.441 |
| | 336 | **0.386** | **0.402** | 0.404 | 0.402 | 0.426 | 0.420 | 0.394 | 0.402 | 0.417 | 0.422 | 0.405 | 0.412 | 0.403 | 0.411 | 0.399 | 0.410 | 0.532 | 0.515 | 0.410 | 0.411 | 0.415 | 0.415 | 0.445 | 0.459 |
| | 720 | **0.452** | **0.437** | 0.463 | 0.437 | 0.491 | 0.459 | 0.460 | 0.442 | 0.481 | 0.458 | 0.466 | 0.447 | 0.461 | 0.442 | 0.454 | 0.439 | 0.666 | 0.589 | 0.478 | 0.450 | 0.473 | 0.451 | 0.543 | 0.490 |
| | Avg. | **0.377** | 0.393 | 0.390 | 0.393 | 0.407 | 0.410 | 0.386 | 0.397 | 0.398 | 0.411 | 0.393 | 0.403 | 0.393 | 0.404 | 0.387 | 0.400 | 0.513 | 0.496 | 0.400 | 0.406 | 0.404 | 0.408 | 0.448 | 0.452 |
| ETTm2 | 96 | **0.169** | **0.255** | 0.174 | 0.255 | 0.180 | 0.264 | 0.176 | 0.257 | 0.177 | 0.262 | 0.180 | 0.261 | 0.176 | 0.266 | 0.175 | 0.259 | 0.287 | 0.366 | 0.187 | 0.267 | 0.193 | 0.293 | 0.203 | 0.287 |
| | 192 | **0.235** | **0.299** | 0.243 | 0.302 | 0.250 | 0.309 | 0.243 | 0.303 | 0.247 | 0.307 | 0.246 | 0.306 | 0.240 | 0.307 | 0.241 | 0.302 | 0.414 | 0.492 | 0.249 | 0.309 | 0.284 | 0.361 | 0.269 | 0.328 |
| | 336 | **0.293** | **0.336** | 0.304 | 0.341 | 0.311 | 0.348 | 0.303 | 0.341 | 0.312 | 0.346 | 0.319 | 0.352 | 0.304 | 0.345 | 0.305 | 0.343 | 0.597 | 0.542 | 0.321 | 0.351 | 0.382 | 0.429 | 0.325 | 0.366 |
| | 720 | **0.390** | **0.393** | 0.399 | 0.397 | 0.412 | 0.407 | 0.400 | 0.398 | 0.414 | 0.403 | 0.405 | 0.401 | 0.406 | 0.400 | 0.402 | 0.400 | 1.730 | 1.042 | 0.408 | 0.403 | 0.558 | 0.525 | 0.421 | 0.415 |
| | Avg. | **0.272** | **0.321** | 0.280 | 0.324 | 0.288 | 0.332 | 0.281 | 0.325 | 0.288 | 0.330 | 0.287 | 0.330 | 0.282 | 0.330 | 0.281 | 0.326 | 0.757 | 0.610 | 0.291 | 0.333 | 0.354 | 0.402 | 0.305 | 0.349 |
| ETTh1 | 96 | **0.370** | 0.394 | 0.377 | **0.393** | 0.386 | 0.405 | 0.377 | 0.394 | 0.390 | 0.411 | 0.381 | 0.399 | 0.382 | 0.398 | 0.414 | 0.419 | 0.423 | 0.448 | 0.384 | 0.402 | 0.397 | 0.412 | 0.376 | 0.419 |
| | 192 | **0.413** | **0.420** | 0.429 | 0.425 | 0.441 | 0.436 | 0.424 | 0.422 | 0.442 | 0.442 | 0.435 | 0.431 | 0.427 | 0.425 | 0.460 | 0.445 | 0.471 | 0.474 | 0.436 | 0.429 | 0.446 | 0.441 | 0.420 | 0.448 |
| | 336 | **0.450** | **0.440** | 0.471 | 0.446 | 0.487 | 0.458 | 0.459 | 0.442 | 0.480 | 0.468 | 0.480 | 0.452 | 0.465 | 0.445 | 0.501 | 0.466 | 0.570 | 0.546 | 0.491 | 0.469 | 0.489 | 0.467 | 0.459 | 0.465 |
| | 720 | **0.448** | **0.457** | 0.496 | 0.480 | 0.503 | 0.491 | 0.463 | 0.459 | 0.494 | 0.488 | 0.499 | 0.488 | 0.472 | 0.468 | 0.500 | 0.488 | 0.653 | 0.621 | 0.521 | 0.500 | 0.513 | 0.510 | 0.506 | 0.507 |
| | Avg. | **0.420** | **0.428** | 0.443 | 0.436 | 0.454 | 0.447 | 0.431 | 0.429 | 0.452 | 0.452 | 0.449 | 0.442 | 0.437 | 0.434 | 0.469 | 0.454 | 0.529 | 0.522 | 0.458 | 0.450 | 0.461 | 0.457 | 0.440 | 0.460 |
| ETTh2 | 96 | **0.283** | **0.337** | 0.296 | 0.345 | 0.297 | 0.349 | 0.292 | 0.343 | 0.328 | 0.371 | 0.297 | 0.347 | 0.309 | 0.359 | 0.302 | 0.348 | 0.745 | 0.584 | 0.340 | 0.374 | 0.340 | 0.394 | 0.358 | 0.397 |
| | 192 | **0.362** | 0.392 | 0.368 | 0.389 | 0.380 | 0.400 | 0.367 | **0.389** | 0.402 | 0.414 | 0.373 | 0.394 | 0.390 | 0.406 | 0.388 | 0.400 | 0.877 | 0.656 | 0.402 | 0.414 | 0.482 | 0.479 | 0.429 | 0.439 |
| | 336 | **0.404** | **0.424** | 0.411 | 0.422 | 0.428 | 0.432 | 0.412 | 0.424 | 0.435 | 0.443 | 0.410 | 0.426 | 0.426 | 0.444 | 0.426 | 0.433 | 1.043 | 0.731 | 0.452 | 0.452 | 0.591 | 0.541 | 0.496 | 0.487 |
| | 720 | **0.407** | **0.433** | 0.412 | 0.434 | 0.427 | 0.445 | 0.419 | 0.438 | 0.417 | 0.441 | 0.411 | 0.433 | 0.445 | 0.464 | 0.431 | 0.446 | 1.104 | 0.763 | 0.462 | 0.468 | 0.839 | 0.661 | 0.463 | 0.474 |
| | Avg. | **0.364** | **0.397** | 0.372 | 0.397 | 0.383 | 0.407 | 0.373 | 0.399 | 0.396 | 0.417 | 0.385 | 0.408 | 0.373 | 0.400 | 0.387 | 0.407 | 0.942 | 0.684 | 0.414 | 0.427 | 0.563 | 0.519 | 0.437 | 0.449 |
| Weather | 96 | **0.153** | **0.199** | 0.163 | 0.202 | 0.174 | 0.214 | 0.156 | 0.202 | 0.163 | 0.212 | 0.166 | 0.208 | 0.159 | 0.218 | 0.177 | 0.218 | 0.158 | 0.230 | 0.172 | 0.220 | 0.195 | 0.252 | 0.217 | 0.296 |
| | 192 | **0.202** | **0.246** | 0.218 | 0.252 | 0.221 | 0.254 | 0.207 | 0.250 | 0.212 | 0.254 | 0.217 | 0.253 | 0.211 | 0.266 | 0.225 | 0.259 | 0.206 | 0.277 | 0.219 | 0.261 | 0.237 | 0.295 | 0.276 | 0.336 |
| | 336 | **0.260** | **0.289** | 0.274 | 0.294 | 0.278 | 0.296 | 0.262 | 0.291 | 0.272 | 0.299 | 0.282 | 0.300 | 0.267 | 0.310 | 0.278 | 0.297 | 0.272 | 0.335 | 0.280 | 0.306 | 0.282 | 0.331 | 0.339 | 0.380 |
| | 720 | **0.342** | **0.341** | 0.349 | 0.343 | 0.358 | 0.349 | 0.343 | 0.343 | 0.350 | 0.348 | 0.356 | 0.351 | 0.352 | 0.362 | 0.354 | 0.348 | 0.398 | 0.418 | 0.365 | 0.359 | 0.345 | 0.382 | 0.403 | 0.428 |
| | Avg. | **0.239** | **0.269** | 0.251 | 0.273 | 0.258 | 0.279 | 0.242 | 0.272 | 0.249 | 0.278 | 0.255 | 0.278 | 0.247 | 0.289 | 0.259 | 0.281 | 0.259 | 0.315 | 0.259 | 0.287 | 0.265 | 0.315 | 0.309 | 0.360 |
| Electricity | 96 | **0.133** | **0.230** | 0.145 | 0.233 | 0.148 | 0.240 | 0.141 | 0.235 | 0.165 | 0.274 | 0.143 | 0.233 | 0.173 | 0.275 | 0.195 | 0.285 | 0.219 | 0.314 | 0.168 | 0.272 | 0.210 | 0.302 | 0.193 | 0.308 |
| | 192 | **0.154** | **0.248** | 0.163 | 0.248 | 0.162 | 0.253 | 0.159 | 0.252 | 0.184 | 0.292 | 0.158 | 0.248 | 0.195 | 0.288 | 0.199 | 0.289 | 0.231 | 0.322 | 0.184 | 0.289 | 0.210 | 0.305 | 0.201 | 0.315 |
| | 336 | **0.162** | **0.261** | 0.175 | 0.262 | 0.178 | 0.269 | 0.173 | 0.268 | 0.198 | 0.302 | 0.178 | 0.269 | 0.206 | 0.300 | 0.215 | 0.305 | 0.246 | 0.337 | 0.198 | 0.300 | 0.223 | 0.319 | 0.214 | 0.329 |
| | 720 | **0.184** | **0.284** | 0.204 | 0.291 | 0.225 | 0.317 | 0.201 | 0.295 | 0.231 | 0.332 | 0.218 | 0.305 | 0.231 | 0.335 | 0.256 | 0.337 | 0.280 | 0.363 | 0.220 | 0.320 | 0.258 | 0.350 | 0.246 | 0.355 |
| | Avg. | **0.158** | **0.256** | 0.172 | 0.258 | 0.178 | 0.270 | 0.169 | 0.263 | 0.194 | 0.300 | 0.174 | 0.264 | 0.201 | 0.300 | 0.216 | 0.304 | 0.244 | 0.334 | 0.193 | 0.295 | 0.225 | 0.319 | 0.214 | 0.327 |
| Traffic | 96 | **0.375** | **0.251** | 0.407 | 0.252 | 0.395 | 0.268 | 0.426 | 0.276 | 0.605 | 0.344 | 0.376 | 0.251 | 0.570 | 0.310 | 0.544 | 0.359 | 0.522 | 0.290 | 0.593 | 0.321 | 0.650 | 0.396 | 0.587 | 0.366 |
| | 192 | **0.395** | 0.262 | 0.431 | 0.262 | 0.417 | 0.276 | 0.458 | 0.289 | 0.613 | 0.359 | 0.398 | **0.261** | 0.577 | 0.321 | 0.540 | 0.354 | 0.530 | 0.293 | 0.617 | 0.336 | 0.598 | 0.370 | 0.604 | 0.373 |
| | 336 | **0.414** | 0.271 | 0.456 | 0.269 | 0.433 | 0.283 | 0.486 | 0.297 | 0.642 | 0.376 | 0.415 | **0.269** | 0.588 | 0.324 | 0.551 | 0.358 | 0.558 | 0.305 | 0.629 | 0.336 | 0.605 | 0.373 | 0.621 | 0.383 |
| | 720 | **0.445** | 0.289 | 0.509 | 0.292 | 0.467 | 0.302 | 0.498 | 0.313 | 0.702 | 0.401 | 0.447 | **0.287** | 0.597 | 0.337 | 0.586 | 0.375 | 0.589 | 0.328 | 0.640 | 0.350 | 0.645 | 0.394 | 0.626 | 0.382 |
| | Avg. | **0.407** | 0.268 | 0.451 | 0.269 | 0.428 | 0.282 | 0.467 | 0.294 | 0.641 | 0.370 | 0.409 | **0.267** | 0.583 | 0.323 | 0.555 | 0.362 | 0.550 | 0.304 | 0.620 | 0.336 | 0.625 | 0.383 | 0.610 | 0.376 |
| Solar-Energy | 96 | **0.193** | 0.223 | 0.200 | **0.207** | 0.203 | 0.237 | 0.197 | 0.241 | 0.208 | 0.243 | 0.200 | 0.230 | 0.283 | 0.353 | 0.234 | 0.286 | 0.310 | 0.331 | 0.250 | 0.292 | 0.290 | 0.378 | 0.242 | 0.342 |
| | 192 | **0.226** | 0.249 | 0.228 | **0.233** | 0.233 | 0.261 | 0.231 | 0.263 | 0.258 | 0.281 | 0.229 | 0.253 | 0.316 | 0.374 | 0.267 | 0.310 | 0.734 | 0.725 | 0.296 | 0.318 | 0.320 | 0.398 | 0.285 | 0.380 |
| | 336 | **0.235** | 0.261 | 0.262 | **0.244** | 0.248 | 0.273 | 0.241 | 0.268 | 0.293 | 0.311 | 0.243 | 0.269 | 0.347 | 0.393 | 0.290 | 0.315 | 0.750 | 0.735 | 0.319 | 0.330 | 0.353 | 0.415 | 0.282 | 0.376 |
| | 720 | **0.239** | 0.268 | 0.258 | **0.249** | 0.249 | 0.275 | 0.250 | 0.281 | 0.290 | 0.315 | 0.245 | 0.272 | 0.348 | 0.389 | 0.289 | 0.317 | 0.769 | 0.765 | 0.338 | 0.337 | 0.357 | 0.413 | 0.357 | 0.427 |
| | Avg. | **0.223** | 0.250 | 0.237 | **0.233** | 0.233 | 0.262 | 0.230 | 0.264 | 0.262 | 0.288 | 0.229 | 0.256 | 0.324 | 0.377 | 0.270 | 0.307 | 0.641 | 0.639 | 0.301 | 0.319 | 0.330 | 0.401 | 0.292 | 0.381 |

*Table 8.* Full results of long-term forecasting. The input sequence length $L$ is set to 96 for all baselines. All results are averaged across four different forecasting horizon: $T \in \{96, 192, 336, 720\}$. The best and second-best results are highlighted in **bold** and underlined, respectively.

| Models | TimeFilter (Ours) | | DUET (2024b) | | CCM (2024) | | PDF (2024) | | ModernTCN (2024) | | iTransformer (2024b) | | CATS (2024) | | PatchTST (2023) | | Crossformer (2023) | | TimesNet (2023) | | DLinear (2023) | |
|---|---|---|---|---|---|---|---|---|---|---|---|---|---|---|---|---|---|---|---|---|---|---|
| Metric | MSE | MAE | MSE | MAE | MSE | MAE | MSE | MAE | MSE | MAE | MSE | MAE | MSE | MAE | MSE | MAE | MSE | MAE | MSE | MAE | MSE | MAE |
| Weather 96 | **0.141** | 0.193 | 0.146 | **0.191** | 0.147 | 0.197 | 0.147 | 0.194 | 0.149 | 0.200 | 0.157 | 0.207 | 0.161 | 0.207 | 0.149 | 0.198 | 0.143 | 0.210 | 0.168 | 0.214 | 0.170 | 0.230 |
| Weather 192 | **0.184** | 0.234 | 0.188 | **0.231** | 0.191 | 0.238 | 0.192 | 0.239 | 0.196 | 0.245 | 0.200 | 0.248 | 0.208 | 0.250 | 0.194 | 0.241 | 0.198 | 0.260 | 0.219 | 0.262 | 0.216 | 0.273 |
| Weather 336 | **0.234** | 0.276 | **0.234** | 0.268 | 0.245 | 0.285 | 0.244 | 0.279 | 0.238 | 0.277 | 0.252 | 0.287 | 0.264 | 0.290 | 0.245 | 0.282 | 0.258 | 0.314 | 0.278 | 0.302 | 0.258 | 0.307 |
| Weather 720 | **0.305** | 0.327 | **0.305** | 0.319 | 0.316 | 0.333 | 0.318 | 0.330 | 0.314 | 0.334 | 0.320 | 0.336 | 0.342 | 0.341 | 0.314 | 0.334 | 0.335 | 0.385 | 0.353 | 0.351 | 0.323 | 0.362 |
| Weather Avg. | **0.216** | 0.258 | 0.218 | **0.252** | 0.225 | 0.263 | 0.225 | 0.261 | 0.224 | 0.264 | 0.232 | 0.270 | 0.244 | 0.272 | 0.226 | 0.264 | 0.234 | 0.292 | 0.255 | 0.282 | 0.242 | 0.293 |
| Electricity 96 | **0.126** | 0.220 | 0.128 | **0.219** | 0.136 | 0.231 | **0.126** | 0.220 | 0.129 | 0.226 | 0.134 | 0.230 | 0.149 | 0.237 | 0.129 | 0.222 | 0.135 | 0.237 | 0.168 | 0.272 | 0.140 | 0.237 |
| Electricity 192 | **0.143** | 0.237 | 0.146 | **0.236** | 0.153 | 0.248 | 0.145 | 0.237 | **0.143** | 0.239 | 0.154 | 0.250 | 0.163 | 0.250 | 0.147 | 0.240 | 0.160 | 0.262 | 0.184 | 0.289 | 0.152 | 0.249 |
| Electricity 336 | **0.153** | **0.252** | 0.163 | 0.254 | 0.168 | 0.262 | 0.159 | 0.255 | 0.161 | 0.259 | 0.169 | 0.265 | 0.180 | 0.268 | 0.163 | 0.259 | 0.182 | 0.282 | 0.198 | 0.300 | 0.169 | 0.267 |
| Electricity 720 | **0.177** | **0.275** | 0.195 | 0.282 | 0.210 | 0.301 | 0.194 | 0.287 | 0.191 | 0.286 | 0.194 | 0.288 | 0.219 | 0.302 | 0.197 | 0.290 | 0.246 | 0.337 | 0.220 | 0.320 | 0.203 | 0.301 |
| Electricity Avg. | **0.150** | **0.246** | 0.158 | 0.248 | 0.167 | 0.261 | 0.156 | 0.250 | 0.156 | 0.253 | 0.163 | 0.258 | 0.178 | 0.264 | 0.159 | 0.253 | 0.181 | 0.279 | 0.192 | 0.295 | 0.166 | 0.264 |
| Traffic 96 | **0.332** | **0.238** | 0.360 | **0.238** | 0.357 | 0.246 | 0.350 | 0.239 | 0.368 | 0.253 | 0.363 | 0.265 | 0.421 | 0.270 | 0.360 | 0.249 | 0.512 | 0.282 | 0.593 | 0.321 | 0.410 | 0.282 |
| Traffic 192 | **0.348** | **0.246** | 0.384 | 0.250 | 0.379 | 0.254 | 0.363 | 0.247 | 0.379 | 0.261 | 0.384 | 0.273 | 0.436 | 0.275 | 0.379 | 0.256 | 0.501 | 0.273 | 0.617 | 0.336 | 0.423 | 0.287 |
| Traffic 336 | **0.365** | 0.256 | 0.395 | 0.260 | 0.389 | **0.255** | 0.376 | 0.258 | 0.397 | 0.270 | 0.396 | 0.277 | 0.453 | 0.284 | 0.392 | 0.264 | 0.507 | 0.279 | 0.629 | 0.336 | 0.436 | 0.296 |
| Traffic 720 | **0.396** | **0.275** | 0.435 | 0.278 | 0.430 | 0.281 | 0.419 | 0.279 | 0.440 | 0.296 | 0.445 | 0.308 | 0.484 | 0.303 | 0.432 | 0.286 | 0.571 | 0.301 | 0.640 | 0.350 | 0.466 | 0.315 |
| Traffic Avg. | **0.360** | **0.254** | 0.394 | 0.257 | 0.389 | 0.259 | 0.377 | 0.256 | 0.396 | 0.270 | 0.397 | 0.281 | 0.449 | 0.283 | 0.391 | 0.264 | 0.523 | 0.284 | 0.620 | 0.336 | 0.434 | 0.295 |
| Climate 96 | **1.030** | **0.492** | 1.090 | 0.510 | 1.103 | 0.511 | 1.116 | 0.530 | 1.100 | 0.514 | 1.044 | 0.494 | 1.157 | 0.535 | 1.354 | 0.519 | 1.083 | 0.508 | 1.274 | 0.524 | 1.108 | 0.536 |
| Climate 192 | **1.039** | **0.489** | 1.098 | 0.511 | 1.118 | 0.510 | 1.362 | 0.570 | 1.125 | 0.523 | 1.055 | 0.496 | 1.073 | 0.507 | 1.388 | 0.572 | 1.115 | 0.532 | 1.274 | 0.541 | 1.122 | 0.536 |
| Climate 336 | **1.056** | **0.489** | 1.120 | 0.515 | 1.130 | 0.513 | 1.396 | 0.568 | 1.153 | 0.530 | 1.069 | 0.495 | 1.072 | 0.495 | 1.410 | 0.581 | 1.246 | 0.575 | 1.285 | 0.584 | 1.146 | 0.545 |
| Climate 720 | **1.061** | **0.490** | 1.103 | 0.498 | 1.143 | 0.515 | 1.386 | 0.552 | 1.243 | 0.599 | 1.073 | 0.497 | 1.078 | 0.498 | 1.395 | 0.575 | 1.236 | 0.584 | 1.347 | 0.626 | 1.158 | 0.545 |
| Climate Avg. | **1.047** | **0.490** | 1.103 | 0.509 | 1.123 | 0.512 | 1.315 | 0.555 | 1.155 | 0.542 | 1.060 | 0.496 | 1.095 | 0.509 | 1.387 | 0.562 | 1.170 | 0.550 | 1.295 | 0.569 | 1.134 | 0.541 |

*Table 9.* Full results of long-term forecasting. The input length $L$ is searched from $\{192, 336, 512, 720\}$ for optimal horizon in the scaling law of TSF (Shi et al., 2024). All results are averaged across four different forecasting horizon: $T \in \{96, 192, 336, 720\}$. The best and second-best results are highlighted in **bold** and underlined, respectively.

| Models | TimeFilter (Ours) | | DUET (2024b) | | iTransformer (2024b) | | Leddam (2024) | | SOFTS (2024a) | | PatchTST (2023) | | Crossformer (2023) | | TimesNet (2023) | | TiDE (2023) | | DLinear (2023) | | SCINet (2022) | | FEDformer (2022) | |
|---|---|---|---|---|---|---|---|---|---|---|---|---|---|---|---|---|---|---|---|---|---|---|---|---|
| Metric | MSE | MAE | MSE | MAE | MSE | MAE | MSE | MAE | MSE | MAE | MSE | MAE | MSE | MAE | MSE | MAE | MSE | MAE | MSE | MAE | MSE | MAE | MSE | MAE |
| PEMS03 12 | **0.063** | **0.165** | 0.064 | 0.166 | 0.071 | 0.174 | 0.068 | 0.174 | 0.064 | **0.165** | 0.099 | 0.216 | 0.090 | 0.203 | 0.085 | 0.192 | 0.178 | 0.305 | 0.122 | 0.243 | 0.066 | 0.172 | 0.126 | 0.251 |
| PEMS03 24 | **0.079** | **0.185** | 0.081 | 0.186 | 0.093 | 0.201 | 0.094 | 0.202 | 0.083 | 0.188 | 0.142 | 0.259 | 0.121 | 0.240 | 0.118 | 0.223 | 0.257 | 0.371 | 0.201 | 0.317 | 0.085 | 0.198 | 0.149 | 0.275 |
| PEMS03 48 | **0.110** | **0.222** | 0.114 | 0.222 | 0.125 | 0.236 | 0.140 | 0.254 | 0.114 | 0.223 | 0.211 | 0.319 | 0.202 | 0.317 | 0.155 | 0.260 | 0.379 | 0.463 | 0.333 | 0.425 | 0.127 | 0.238 | 0.227 | 0.348 |
| PEMS03 Avg. | **0.084** | **0.191** | 0.086 | 0.192 | 0.096 | 0.204 | 0.101 | 0.210 | 0.087 | 0.192 | 0.151 | 0.265 | 0.138 | 0.253 | 0.119 | 0.271 | 0.271 | 0.380 | 0.219 | 0.295 | 0.093 | 0.203 | 0.167 | 0.291 |
| PEMS04 12 | **0.068** | **0.167** | 0.079 | 0.181 | 0.078 | 0.183 | 0.076 | 0.182 | 0.074 | 0.176 | 0.105 | 0.224 | 0.098 | 0.218 | 0.087 | 0.195 | 0.219 | 0.340 | 0.148 | 0.272 | 0.073 | 0.177 | 0.138 | 0.262 |
| PEMS04 24 | **0.080** | **0.183** | 0.096 | 0.203 | 0.095 | 0.205 | 0.097 | 0.209 | 0.088 | 0.194 | 0.153 | 0.257 | 0.131 | 0.256 | 0.103 | 0.215 | 0.292 | 0.398 | 0.224 | 0.340 | 0.084 | 0.193 | 0.177 | 0.293 |
| PEMS04 48 | 0.101 | 0.209 | 0.114 | 0.226 | 0.120 | 0.233 | 0.132 | 0.249 | 0.110 | 0.219 | 0.229 | 0.339 | 0.205 | 0.326 | 0.136 | 0.250 | 0.409 | 0.478 | 0.335 | 0.437 | **0.099** | 0.211 | 0.270 | 0.368 |
| PEMS04 Avg. | **0.083** | **0.186** | 0.096 | 0.203 | 0.098 | 0.207 | 0.102 | 0.213 | 0.091 | 0.196 | 0.162 | 0.273 | 0.145 | 0.267 | 0.109 | 0.220 | 0.307 | 0.405 | 0.236 | 0.350 | 0.085 | 0.194 | 0.195 | 0.308 |
| PEMS07 12 | **0.055** | **0.150** | 0.060 | 0.156 | 0.067 | 0.165 | 0.066 | 0.164 | 0.057 | 0.152 | 0.095 | 0.207 | 0.094 | 0.200 | 0.082 | 0.181 | 0.173 | 0.304 | 0.115 | 0.242 | 0.068 | 0.171 | 0.109 | 0.225 |
| PEMS07 24 | **0.068** | **0.166** | 0.073 | 0.172 | 0.088 | 0.190 | 0.079 | 0.185 | 0.073 | 0.173 | 0.150 | 0.262 | 0.139 | 0.247 | 0.101 | 0.204 | 0.271 | 0.383 | 0.210 | 0.329 | 0.119 | 0.225 | 0.125 | 0.244 |
| PEMS07 48 | **0.089** | **0.193** | 0.096 | 0.201 | 0.110 | 0.215 | 0.115 | 0.228 | 0.096 | 0.195 | 0.253 | 0.340 | 0.311 | 0.369 | 0.134 | 0.238 | 0.446 | 0.495 | 0.398 | 0.458 | 0.149 | 0.237 | 0.165 | 0.288 |
| PEMS07 Avg. | **0.071** | **0.170** | 0.076 | 0.176 | 0.088 | 0.190 | 0.087 | 0.192 | 0.075 | 0.173 | 0.166 | 0.270 | 0.181 | 0.272 | 0.106 | 0.208 | 0.297 | 0.394 | 0.241 | 0.343 | 0.112 | 0.211 | 0.133 | 0.282 |
| PEMS08 12 | **0.064** | **0.162** | 0.072 | 0.168 | 0.079 | 0.182 | 0.070 | 0.173 | 0.074 | 0.171 | 0.168 | 0.232 | 0.165 | 0.214 | 0.112 | 0.212 | 0.227 | 0.343 | 0.154 | 0.276 | 0.087 | 0.184 | 0.173 | 0.273 |
| PEMS08 24 | **0.079** | **0.182** | 0.093 | 0.191 | 0.115 | 0.219 | 0.091 | 0.200 | 0.104 | 0.201 | 0.224 | 0.281 | 0.215 | 0.260 | 0.141 | 0.238 | 0.318 | 0.409 | 0.248 | 0.353 | 0.122 | 0.221 | 0.210 | 0.310 |
| PEMS08 48 | **0.105** | **0.214** | 0.123 | 0.217 | 0.186 | 0.235 | 0.145 | 0.261 | 0.164 | 0.253 | 0.321 | 0.354 | 0.315 | 0.335 | 0.198 | 0.283 | 0.497 | 0.510 | 0.440 | 0.470 | 0.189 | 0.270 | 0.320 | 0.394 |
| PEMS08 Avg. | **0.083** | **0.186** | 0.096 | 0.192 | 0.127 | 0.212 | 0.102 | 0.211 | 0.114 | 0.208 | 0.238 | 0.289 | 0.232 | 0.270 | 0.150 | 0.244 | 0.347 | 0.421 | 0.281 | 0.366 | 0.133 | 0.225 | 0.234 | 0.326 |

*Table 10.* Full results of short-term forecasting. The input sequence length is set to 96 for all baselines. All results are averaged across four different forecasting horizon: $T \in \{12, 24, 48\}$. The best and second-best results are highlighted in **bold** and underlined, respectively.

| Categories | | Filter Replace | | | | | | | | | | | | | | MoE | | | |
|---|---|---|---|---|---|---|---|---|---|---|---|---|---|---|---|---|---|---|---|---|
| Cases | | TimeFilter | | Top-$K$ | | Random-$K$ | | RegionTop-$K$ | | RegionThre | | C-Filter | | w/o Filter | | Static Routing | | w/o $\mathcal{L}_{\text{imp \& dyn}}$ | |
| Metric | | MSE | MAE | MSE | MAE | MSE | MAE | MSE | MAE | MSE | MAE | MSE | MAE | MSE | MAE | MSE | MAE | MSE | MAE |
| Weather | 96 | **0.153** | **0.199** | 0.158 | 0.203 | 0.158 | 0.204 | 0.156 | 0.202 | 0.156 | 0.202 | 0.155 | 0.202 | 0.159 | 0.204 | 0.156 | 0.203 | 0.156 | 0.203 |
| | 192 | 0.202 | 0.246 | 0.207 | 0.250 | 0.207 | 0.251 | 0.205 | 0.249 | 0.202 | 0.247 | 0.206 | 0.249 | 0.208 | 0.251 | 0.205 | 0.248 | 0.206 | 0.248 |
| | 336 | 0.260 | 0.289 | 0.264 | 0.292 | 0.263 | 0.292 | 0.262 | 0.292 | 0.262 | 0.291 | 0.263 | 0.291 | 0.265 | 0.292 | 0.263 | 0.290 | 0.264 | 0.291 |
| | 720 | 0.342 | 0.341 | 0.344 | 0.343 | 0.342 | 0.341 | 0.342 | 0.342 | 0.341 | 0.341 | 0.343 | 0.342 | 0.345 | 0.345 | 0.343 | 0.342 | 0.344 | 0.344 |
| | *Avg.* | **0.239** | **0.269** | 0.243 | 0.272 | 0.243 | 0.272 | 0.241 | 0.271 | 0.241 | 0.270 | 0.242 | 0.271 | 0.244 | 0.273 | 0.242 | 0.271 | 0.243 | 0.272 |
| Electricity | 96 | 0.133 | 0.230 | 0.138 | 0.235 | 0.137 | 0.235 | 0.139 | 0.235 | 0.138 | 0.234 | 0.137 | 0.234 | 0.139 | 0.237 | 0.137 | 0.234 | 0.136 | 0.233 |
| | 192 | 0.154 | 0.248 | 0.163 | 0.255 | 0.161 | 0.254 | 0.163 | 0.255 | 0.164 | 0.257 | 0.159 | 0.252 | 0.159 | 0.252 | 0.156 | 0.250 | 0.155 | 0.250 |
| | 336 | 0.162 | 0.261 | 0.168 | 0.266 | 0.174 | 0.270 | 0.169 | 0.265 | 0.166 | 0.264 | 0.169 | 0.267 | 0.168 | 0.265 | 0.167 | 0.264 | 0.166 | 0.263 |
| | 720 | 0.184 | 0.284 | 0.200 | 0.297 | 0.200 | 0.297 | 0.199 | 0.295 | 0.220 | 0.314 | 0.214 | 0.308 | 0.200 | 0.298 | 0.195 | 0.290 | 0.193 | 0.288 |
| | *Avg.* | **0.158** | **0.256** | 0.167 | 0.263 | 0.168 | 0.264 | 0.168 | 0.263 | 0.172 | 0.267 | 0.170 | 0.265 | 0.167 | 0.263 | 0.164 | 0.260 | 0.163 | 0.259 |
| Traffic | 96 | 0.375 | 0.251 | 0.379 | 0.253 | 0.378 | 0.253 | 0.377 | 0.251 | 0.378 | 0.251 | 0.379 | 0.252 | 0.379 | 0.254 | 0.377 | 0.253 | 0.378 | 0.254 |
| | 192 | 0.395 | 0.262 | 0.404 | 0.265 | 0.402 | 0.264 | 0.403 | 0.263 | 0.406 | 0.264 | 0.405 | 0.264 | 0.404 | 0.266 | 0.398 | 0.264 | 0.397 | 0.264 |
| | 336 | 0.414 | 0.271 | 0.420 | 0.274 | 0.418 | 0.273 | 0.416 | 0.272 | 0.417 | 0.275 | 0.418 | 0.274 | 0.422 | 0.275 | 0.417 | 0.273 | 0.417 | 0.272 |
| | 720 | 0.445 | 0.289 | 0.449 | 0.293 | 0.448 | 0.292 | 0.449 | 0.292 | 0.447 | 0.291 | 0.448 | 0.291 | 0.450 | 0.295 | 0.447 | 0.292 | 0.448 | 0.293 |
| | *Avg.* | **0.407** | **0.268** | 0.413 | 0.271 | 0.412 | 0.271 | 0.411 | 0.270 | 0.412 | 0.271 | 0.413 | 0.271 | 0.414 | 0.273 | 0.410 | 0.271 | 0.410 | 0.271 |
| Solar-Energy | 96 | 0.193 | 0.223 | 0.203 | 0.229 | 0.198 | 0.225 | 0.202 | 0.224 | 0.204 | 0.225 | 0.203 | 0.225 | 0.201 | 0.234 | 0.196 | 0.224 | 0.194 | 0.224 |
| | 192 | 0.226 | 0.249 | 0.225 | 0.255 | 0.228 | 0.257 | 0.225 | 0.254 | 0.224 | 0.254 | 0.233 | 0.257 | 0.230 | 0.262 | 0.230 | 0.262 | 0.227 | 0.262 |
| | 336 | 0.235 | 0.261 | 0.249 | 0.270 | 0.244 | 0.269 | 0.253 | 0.271 | 0.254 | 0.274 | 0.251 | 0.273 | 0.249 | 0.272 | 0.242 | 0.269 | 0.242 | 0.270 |
| | 720 | 0.239 | 0.268 | 0.248 | 0.276 | 0.252 | 0.280 | 0.250 | 0.277 | 0.256 | 0.281 | 0.250 | 0.278 | 0.256 | 0.280 | 0.245 | 0.276 | 0.244 | 0.275 |
| | *Avg.* | **0.223** | **0.250** | 0.231 | 0.258 | 0.231 | 0.258 | 0.233 | 0.257 | 0.235 | 0.259 | 0.234 | 0.258 | 0.234 | 0.262 | 0.228 | 0.258 | 0.227 | 0.258 |
| ETTh1 | 96 | 0.370 | 0.394 | 0.379 | 0.399 | 0.377 | 0.396 | 0.379 | 0.398 | 0.379 | 0.398 | 0.377 | 0.399 | 0.379 | 0.398 | 0.374 | 0.397 | 0.374 | 0.394 |
| | 192 | 0.413 | 0.420 | 0.432 | 0.429 | 0.430 | 0.427 | 0.432 | 0.429 | 0.434 | 0.428 | 0.424 | 0.428 | 0.431 | 0.429 | 0.421 | 0.424 | 0.421 | 0.423 |
| | 336 | 0.450 | 0.440 | 0.478 | 0.454 | 0.475 | 0.453 | 0.479 | 0.454 | 0.469 | 0.449 | 0.467 | 0.452 | 0.475 | 0.453 | 0.460 | 0.446 | 0.460 | 0.445 |
| | 720 | 0.448 | 0.457 | 0.474 | 0.468 | 0.482 | 0.473 | 0.475 | 0.469 | 0.466 | 0.467 | 0.457 | 0.463 | 0.481 | 0.471 | 0.465 | 0.467 | 0.465 | 0.467 |
| | *Avg.* | **0.420** | **0.428** | 0.441 | 0.438 | 0.441 | 0.437 | 0.441 | 0.438 | 0.437 | 0.436 | 0.431 | 0.436 | 0.442 | 0.438 | 0.440 | 0.434 | 0.430 | 0.432 |
| ETTm2 | 96 | 0.169 | 0.255 | 0.172 | 0.257 | 0.172 | 0.257 | 0.170 | 0.257 | 0.172 | 0.257 | 0.172 | 0.258 | 0.173 | 0.257 | 0.172 | 0.256 | 0.171 | 0.256 |
| | 192 | 0.235 | 0.299 | 0.237 | 0.300 | 0.238 | 0.301 | 0.233 | 0.299 | 0.237 | 0.299 | 0.235 | 0.300 | 0.237 | 0.300 | 0.236 | 0.298 | 0.235 | 0.298 |
| | 336 | 0.293 | 0.336 | 0.298 | 0.340 | 0.296 | 0.338 | 0.298 | 0.340 | 0.297 | 0.340 | 0.299 | 0.342 | 0.298 | 0.340 | 0.295 | 0.337 | 0.294 | 0.336 |
| | 720 | 0.390 | 0.393 | 0.395 | 0.398 | 0.396 | 0.397 | 0.392 | 0.396 | 0.399 | 0.398 | 0.400 | 0.400 | 0.400 | 0.399 | 0.392 | 0.396 | 0.393 | 0.397 |
| | *Avg.* | **0.272** | **0.321** | 0.276 | 0.324 | 0.276 | 0.323 | 0.273 | 0.323 | 0.276 | 0.324 | 0.277 | 0.325 | 0.277 | 0.324 | 0.274 | 0.322 | 0.273 | 0.322 |
| PEMS08 | 12 | 0.064 | 0.162 | 0.065 | 0.165 | 0.066 | 0.165 | 0.064 | 0.164 | 0.065 | 0.163 | 0.064 | 0.164 | 0.065 | 0.164 | 0.064 | 0.163 | 0.064 | 0.162 |
| | 24 | 0.079 | 0.182 | 0.081 | 0.185 | 0.081 | 0.184 | 0.081 | 0.186 | 0.081 | 0.185 | 0.082 | 0.185 | 0.082 | 0.186 | 0.080 | 0.184 | 0.080 | 0.183 |
| | 48 | 0.105 | 0.214 | 0.112 | 0.224 | 0.108 | 0.217 | 0.110 | 0.219 | 0.108 | 0.219 | 0.111 | 0.222 | 0.110 | 0.223 | 0.108 | 0.216 | 0.108 | 0.217 |
| | *Avg.* | **0.083** | **0.186** | 0.086 | 0.191 | 0.085 | 0.189 | 0.085 | 0.190 | 0.085 | 0.189 | 0.086 | 0.190 | 0.086 | 0.191 | 0.084 | 0.188 | 0.084 | 0.187 |

*Table 11.* Component ablation of TimeFilter.

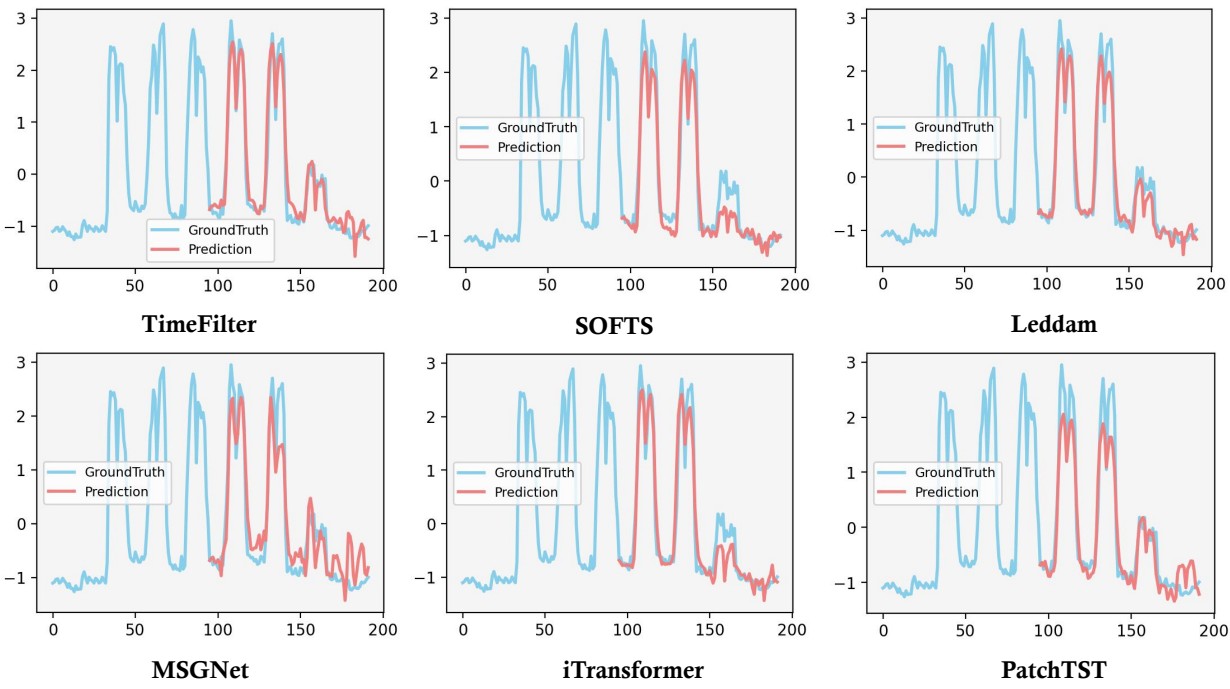

*Figure 7.* Visualization of predictions from different models on the Electricity dataset.

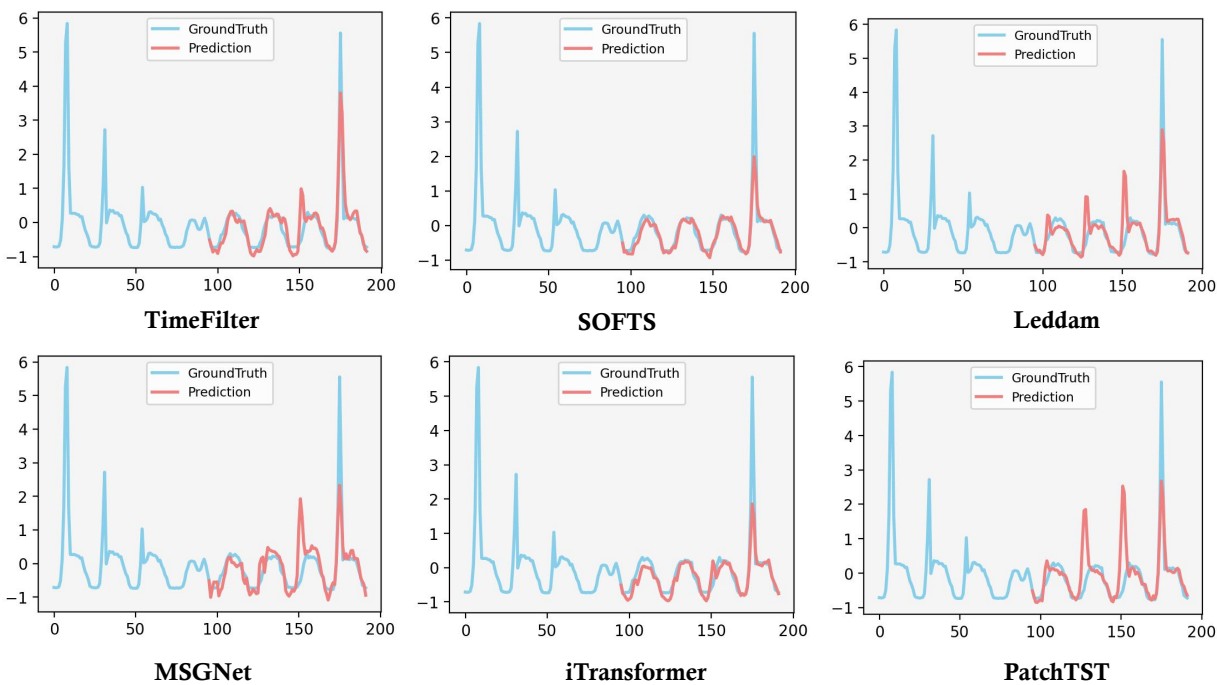

*Figure 8.* Visualization of predictions from different models on the Traffic dataset.

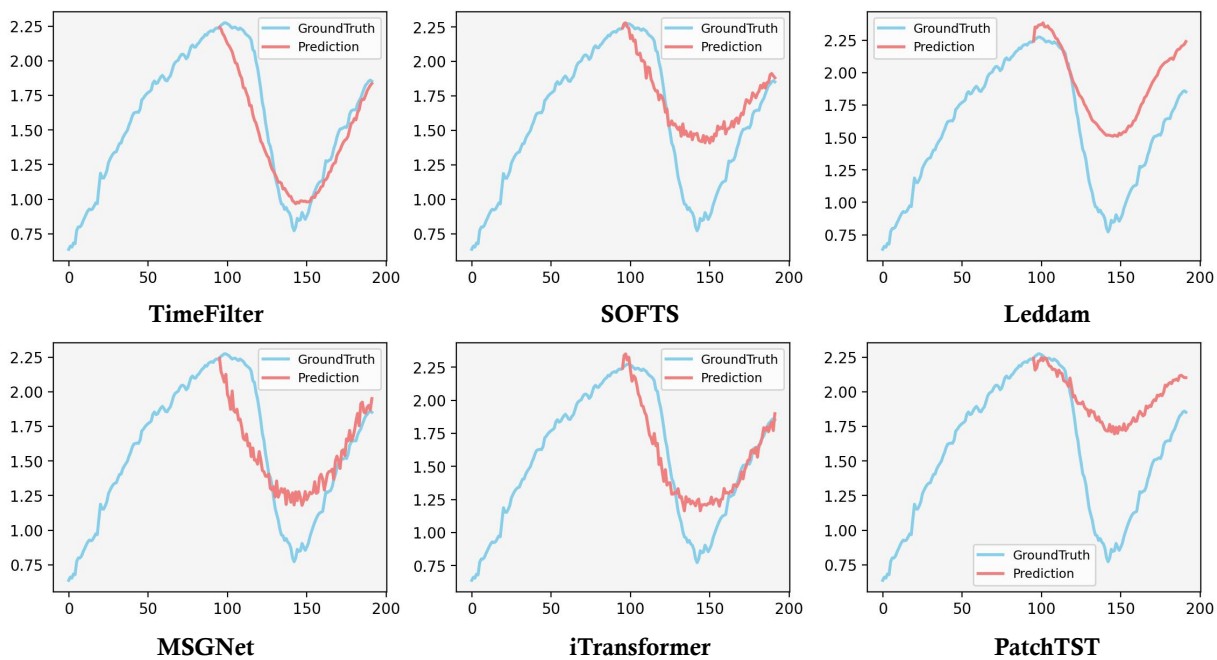

*Figure 9.* Visualization of predictions from different models on the Weather dataset.

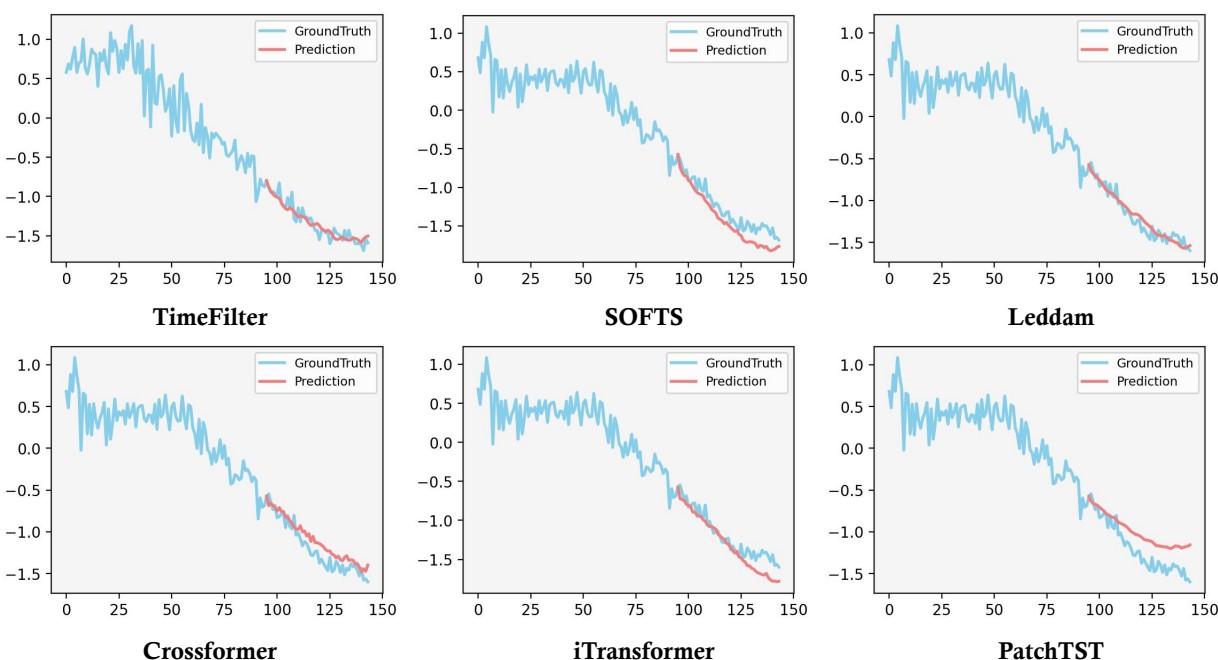

*Figure 10.* Visualization of predictions from different models on the PEMS08 dataset.

