# OpenReview forum: "TimeFilter: Patch-Specific Spatial-Temporal Graph Filtration for Time Series Forecasting"
_ICML.cc/2025/Conference — ICML 2025 poster_

### Official Review · Reviewer_Utah · 2025-03-08

**Overall Recommendation:** 4

**Summary:**

This paper introduces a novel dependency modeling strategy for time series forecasting, distinct from the commonly used channel independent (CI) and channel dependent approaches (CD). The proposed Patch-wise Filtration method balances CI and CD, allowing for fine-grained and dynamic modeling of spatiotemporal relationships between patches while considering computational complexity. The experimental results look convincing, and the ablation studies enumerate several alternative solutions, demonstrating the benefits of TimeFilter's filtering mechanism based on the MoE architecture.

**Claims And Evidence:**

This paper asserts that the relationships between patches become effective or ineffective over time, which is difficult to capture using channel-wise modeling methods. First, this claim is intuitive. Moreover, the example illustration in the Introduction and the experimental figures in the Case Study clearly validate this assertion. Additionally, the visualization of the four modeling approaches highlights the differences between various granularity-based modeling methods.

**Essential References Not Discussed:**

This paper has a very good coverage of the literature.

**Experimental Designs Or Analyses:**

The experiments in this paper are comprehensive, covering both short-term and long-term forecasting tasks, ablation studies, case studies, analysis of training time and memory consumption, as well as the impact of the lookback horizon. The above results look convincing and I cannot find major flaws.

**Methods And Evaluation Criteria:**

This paper introduces a novel dependency modeling approach, offering a fresh perspective to the time series forecasting community. The usage of datasets and metrics follows the standard practices in previous works.

**Other Comments Or Suggestions:**

N.A.

**Other Strengths And Weaknesses:**

Strengths:

1. The proposed framework is a new perspective in time series dependency modeling, which performs well in terms of both forecasting performance and efficiency.
2. This paper conducts extensive experiments.
3. The writing is clear and the figures are pleasing.
4. The paper consistently presents clear intuitions and thoughtful design choices.
5. I especially appreciate the case study in the paper, which draws a conclusion to the continuous debate between CI and CD methods in recent years.

Weaknesser:

1. The theoretical complexity of this method is quadratic, which does not offer the low memory footprint characteristic of linear-based methods.
2. Though not being a deal breaker, the paper lacks a clear description of the hyperparameter tuning process, especially regarding the patch length parameter. In the appendix, the chosen patch lengths vary significantly across different datasets, raising questions about how this hyperparameter is determined.
3. The idea of masking (or pruning) dependencies has existed before, although not as refined or effective as presented in this paper.

**Questions For Authors:**

1. The authors convert each patch into an ego graph and prune it, claiming that this operation can be parallelized without introducing additional complexity. However, I am curious about how exactly this parallelization is achieved, as the paper does not provide a clear explanation.
2. Masking attention can achieve similar effects. Why did the authors opt for a pure GNN architecture instead?

**Relation To Broader Scientific Literature:**

The choice between CI and CD, and how to better model dependencies for temporal representation, is always a hot topic in time series community. The findings and implementation in this paper can be conducive to tasks such as time series forecasting, imputation, classification, and anomaly detection.

**Theoretical Claims:**

There is no theoretical claim in this paper.

---

> ### Author Rebuttal · Authors · 2025-04-01
>
> Thank you for your insightful advice for polishing our manuscript. We have conducted sufficient experiments and analysis to dispel your concerns. The details can be found below.
>
> ## Other Strengths And Weaknesses
>
> ---
> **`W1`: Alalysis of theoretical complexity.**
>
> **`R1`:** We include a detailed comparison of the theoretical complexity of our method and other SOTA models. The Tab. below summarizes the theoretical computational complexity, where C is the number of channels, L is the input lengths, and P is the patch length.
> |TimeFilter|iTransformer|PatchTST|Crossformer|
> |-|-|-|-|
> |$O(C^2·(\frac{L}{P})^2)$|$O(C^2)$|$O(C·(\frac{L}{P})^2)$|$O(\frac{C}{P^2}·L^2)$|
>
> Moreover, theoretical complexity alone cannot fully capture real-world performance due to implementation differences. We tested on one NVIDIA A100 GPUs, measuring training (1 epoch) and inference times for three datasets of increasing size, with results averaged over 5 runs in tha table below.
> |||TimeFilter|iTransformer|PatchTST|Crossformer|
> |-|-|-|-|-|-|
> |Weather ($C=21$)|Training|19s|23s|41s|74s|
> ||Inference|4s|6s|9s|11s|
> |Electricity ($C=321$)|Training|91s|74s|280s|248s|
> ||Inference|14s|11s|36s|34s|
> |Traffic ($C=862$)|Training|77s|101s|352s|360s|
> ||Inference|14s|18s|66s|81s|
>
> ---
> **`W2`: Description of the hyperparameter tuning process, especially the patch length parameter.**
>
> **`R2`:** We conducted an extensive hyperparameter search covering learning rates from $5\times 10^{-4}$ to $10^{-3}$, encoder layers from $1$ to $3$, $d_{model}$ values from $32$ to $512$, training epochs from $10$ to $30$, and patch lengths from $2$ to $96$.
>
> In particular, the patch length $P$ has a significant impact on the results as it determines the contextual semantic information captured. We chose $P$ based on the characteristics of the dataset, experimenting with values of $L, \frac{L}{2}, \frac{L}{3}$, etc. This data-driven, result-oriented tuning ensures optimal performance. For transparency, we'll describe this process in detail in the appendix.
>
> ---
> **`W3`: Difference with other masking (or pruning) dependencies methods.**
>
> **`R3`:** While previous work has explored dependency masking, our approach differs itself by filtering based on dependency types rather than binary causal relationships. Traditional methods often mask dependencies by assuming fixed causal relationships, which can be misleading due to spurious correlations in short-term dependencies.
> Our innovation lies in recognizing that different domains require distinct dependency types. For example: Traffic flow prediction benefits from spatial dependencies (e.g., interactions between adjacent road segments); financial markets often require filtering out short-term noise while preserving long-term trends; electricity demand forecasting relies heavily on periodic dependencies tied to daily/seasonal cycles.
> The significant performance gains in our experiments validate the effectiveness of this type of dependency filtering and demonstrate its superiority over generic causal masking strategies.
>
> ## Questions For Authors
>
> ---
> **`Q1`: Details of parallel pruning.**
>
> **`R1`:** Sorry for the confusion. For each patch i, its ego graph corresponds to a row in the overall adjacency matrix M. Pruning is achieved by directly modifying M. For the three filters, we pre-generate masks. Then, by element-wise multiplication of M with the selected mask via Einstein summation, we efficiently parallelize the filtering without any additional complexity. We'll clarify this in the revised manuscript.
>
> ---
> **`Q2`: Why choose a pure GNN architecture rather than attention?**
>
> **`R2`:** We appreciate this insightful technical question. The choice of a GNN-based architecture over masked self-attention is motivated by both computational efficiency and inherent advantages in modeling temporal dependencies. GNNs offer lower complexity at $O(Qm)$ where m, pruned to O(kn), is the number of edges, compared to Transformers' quadratic $O(n^2)$ complexity (Q layers, m edges, n nodes). GNNs are inherently adept at modeling instance relationships through explicit graph structures, preserving inductive bias and offering better interpretability than the often opaque attention weights. We'll ensure these advantages are clearly articulated in the revised manuscript.
>
> We hope our response addresses your concerns.

---

### Official Review · Reviewer_1eke · 2025-03-09

**Overall Recommendation:** 4

**Summary:**

This paper proposes a Patch-wise filtering modeling approach to select important dependencies and remove irrelevant noisy relationships. It integrates the benefits of CI and CD strategies and offers a more fine-grained and adaptive consideration of dynamically evolving dependencies over time compared to the CC strategy. The paper conducts extensive experiments, demonstrating significant improvements across various lookback lengths. Additionally, detailed ablation studies are provided. Overall, the paper is of good quality.

**Claims And Evidence:**

The paper substantiates its claim of dynamic, fine-grained relationships between patches by visualizing the dependency graphs and the distribution of gating mechanisms during the modeling process.

**Essential References Not Discussed:**

I think there are no essential references that have not been discussed.

**Experimental Designs Or Analyses:**

The experimental design in this paper is more interesting compared to other studies. In addition to using fixed lengths, the authors introduced extended experiments from the perspective of the scaling law of TSF. The ablation studies compared TimeFilter with other intuitive filtering methods to demonstrate the rationality of their design. They also provide visualisations to illustrate the performance improvements brought by the proposed modules. The code is provided, too.

**Methods And Evaluation Criteria:**

The proposed method, TimeFilter, offers a new and better alternative to the conventional modeling approaches of CI and CD. Like many other papers, the paper also employs commonly used time series forecasting datasets such as ETT, Traffic, and so on, as well as the widely adopted error metrics of MSE and MAE.

**Other Comments Or Suggestions:**

1. In Equation 14, the specific operation of **COMB** in TimeFilter is not clearly explained.

2. In Equation 18, the value of λ1 and λ2 are not specified.

**Other Strengths And Weaknesses:**

Strengths:

1. The paper is well-motivated, as the challenge of effectively modeling both temporal and inter-channel relationships.

2. This paper is well written. The notations are clear.

3. The experiments are comprehensive, and the compared baselines are up-to-date.

Weaknesses:

1. Some modules exhibit limited novelty. For instance, the idea of Dynamic Expert Allocation is not first proposed in this paper.

2. It would be better if the authors also visualize the dependency graphs of the other baselines.

**Questions For Authors:**

Why is the dependency modeling graph of TimeFilter in Figure 2(c) not symmetric along the diagonal? Shouldn’t the relationship between patch i and patch j be filtered simultaneously in both directions, removing the influence of patch i on patch j and vice versa?

**Relation To Broader Scientific Literature:**

In addition to TSF, other domains such as spatio-temporal prediction, speech and acoustics also involve similar fine-grained, dynamically evolving dependencies. This paper offers new insights into leveraging such dependencies.

**Theoretical Claims:**

There are no theoretical claims.

---

> ### Author Rebuttal · Authors · 2025-04-01
>
> Thank you for your insightful advice. Here are responses to your questions:
>
> ## Other Strengths And Weaknesses
>
> **`W1`: Some modules exhibit limited novelty. For instance, the idea of Dynamic Expert Allocation is not first proposed in this paper.**
>
> **`R1`:** We sincerely appreciate your deep expertise in identifying this important aspect. Dynamic MOE's adaptive token-wise expert allocation to address predetermined expert selection constraints, our TimeFilter does indeed build on this previous work[1,2], but we would like to clarify that our paper does not claim this as our primary contribution. Our key innovation lies in fine-grained dependency modeling beyond CD/CI, which can be realized through various mechanisms. Dynamic MOE serves as our implementation choice, but alternative approaches like attention-gated dependency routers or hierarchical graph convolutions with adaptive receptive fields could also instantiate our core paradigm. The novelty lies in the proposed dependency disentanglement framework rather than the specific routing mechanism.
>
> [1] Harder Tasks Need More Experts: Dynamic Routing in MoE Models.
>
> [2] Dynamic Mixture of Experts: An Auto-Tuning Approach for Efficient Transformer Models.
>
> ---
> **`W2`: It would be better if the authors also visualize the dependency graphs of the other baselines.**
>
> **`R2`:** We've visualized PatchTST's and iTransformer's attention on Weather dataset in https://i.postimg.cc/Jn42zB0b/rebuttal-fig.png. TimeFilter achieves more precise dependency modeling by selectively focusing on specific types of relationships, unlike PatchTST's narrow focus on mutation points and iTransformer's overly generalized attention across periods. This selective approach allows TimeFilter to better identify meaningful interactions while filtering out noise.
>
> ## Other Comments Or Suggestions
>
> ---
> **`C1`: Explanation of the **COMB** operation in TimeFilter.**
>
> **`R1`:** Sorry for the confusion. COMB refers to the combination of ego and neighbour representations. We implement COMB using a simple FFN, similar to GCN [1]. We'll clarify this in the revised manuscript for better understanding.
>
> [1] Semi-Supervised Classification with Graph Convolutional Networks.
>
> ---
> **`C2`:** Hyper-parameters explanation.
>
> **`R2`:** In Equation 18, $\lambda_1$ and $\lambda_2$ are scaling factors for the loss functions. In our experiments, we set $\lambda_1$ to 0.05 and $\lambda_2$ to 0.005. These values ensure the three loss functions are balanced, facilitating gradient optimization.
>
> ## Questions For Authors
>
> ---
> **`Q1`: Why is the dependency graph of TimeFilter in Figure 2(c) not symmetric along the diagonal?**
>
> **`R1`:** The dependency modeling graph in Figure 2(c) is not symmetric because each patch has its own customized dependency relationships. Specifically, the filtering process for patch i's ego graph is independent of patch j's, and the selected filters may differ between them. This asymmetry is intuitive in real-world scenarios. For example, in financial systems, macroeconomic policy factors can directly influence price-volume factors (e.g., a positive policy change boosts market activity), but the reverse influence (e.g., price-volume changes trigger policy adjustments) is often weaker or absent. Thus, asymmetric dependencies naturally arise in such contexts.
>
> ---
> Once again, we are deeply grateful for your recognition of our work and your constructive feedback. If you have any further questions or suggestions, we would be more than happy to address them.

---

### Official Review · Reviewer_us7m · 2025-03-14

**Overall Recommendation:** 3

**Summary:**

This paper introduces a novel approach that addresses the limitations of channel-independent (CI) and channel-dependent (CD) and channel-claustering (CC) strategies. The proposed TimeFilter transition from previous coarse-grained, channel-wise clustering approaches
to a finer-grained, patch-wise partitioning strategy. Specifically, TimeFilter constructs a spatial-temporal graph based on patch-level distances by k-Nearest Neighbor methods. Moreover, a mixture-of-experts mechanism is applied to dynamically route and filter dependencies for each time patch. Adaptive Graph Learning (AGL) Module updates the time series embedding through neighborhood aggregation and performs forecasting. Extensive experiments show that TimeFilter generally outperforms other baselines, validating the effectiveness of the proposed TimeFilter framework.

**Claims And Evidence:**

-  The paper claims appropriate dependency modeling strategies, However, there is no direct quantitative breakdown of how much each component of the constructed dependency graphs contributes to the performance improvement. There is a noticeable lack of ablation studies on the importance of temporal,  spatial, and spatial-temporal subgraphs.
- The paper shows the impact of the filtering component. However, the definition of "irrelevant" is implicit in the model's learning process. Visualizations or statistical analysis of the filtered connections could be beneficial.

**Essential References Not Discussed:**

N/A

**Experimental Designs Or Analyses:**

The paper uses standard benchmark datasets for time series forecasting and includes a good range of baseline methods.

**Methods And Evaluation Criteria:**

The proposed TimeFilter and the decomposition of temporal, spatial, and spatial-temporal subgraphs are reasonable for cross-channel time series modeling. Discussions exploring real-world scenarios and specific use cases where these three graph types prove necessary would be beneficial.

**Other Comments Or Suggestions:**

N/A

**Other Strengths And Weaknesses:**

Pros:
- The motivation for dependency filtering and graph modeling for cross-channel time series forecasting is clear and reasonable.
- The paper is well-written, clear and easy to follow.
- Codes are available, which promotes the reproducibility of the work.

Cons:
- The sensitivity of hyper-parameters (e.g., number of nodes $n$, threshold $p$) is unclear.  There is a lack of detailed ablation study on these necessary hyper-parameters.
- The claim of "Paradigm Transformation" might be seen as exaggerated. While the method is somewhat novel, it builds upon existing research in time series channel strategies and graph neural networks.

**Questions For Authors:**

- What is the computational cost of graph construction, compared with other modules in TimeFilter?
- In Figure 4, the scores for the 3 filter for each dependency subgraph are close to 0. Does that mean $m=2$ is sufficient for TimeFilter modeling?
-  What is the complexity of each component in the proposed TimeFilter? There is no formal analysis of the model's complexity.

**Relation To Broader Scientific Literature:**

The paper clearly articulates the limitations of existing CI, CD, and CC methods and unifies these channel strategies in Figure 2. TimeFilter achieves the best trade-off between complete separated and complete collaborative channel dependencies.

**Theoretical Claims:**

There is no proof for theoretical claims in the paper.

---

> ### Author Rebuttal · Authors · 2025-04-01
>
> Thanks for your valuable comments. Here are detailed responses to your questions:
>
> ## Claims And Evidence
> **E1:** Claims of appropriate dependency modeling strategies.
>
> **R1:** We conducted ablation studies in the table below with variants: tem.-only (T), spa.-only (S), spa.-tem.-only (ST), and their combinations without filtering. The results show that no single dependency combination outperforms others across datasets. The context-aware filtering mechanism reduces prediction errors by adaptively suppressing noise. This confirms that TimeFilter’s performance results from the principled fusion of complementary dependencies, not isolated components. We will expand this analysis in revision.
> ||TimeFilter|T|S|ST|T&S|T&ST|S&ST|T&S&ST|
> |-|-|-|-|-|-|-|-|-|
> |Weather|**0.239**/**0.269**|0.242/0.272|0.241/0.272|0.241/0.271|0.241/0.271|0.241/0.271|0.241/0.272|0.243/0.274|
> |ECL|**0.158**/**0.256**|0.168/0.265|0.170/0.269|0.162/0.260|0.171/0.271|0.162/0.260|0.163/0.261|0.166/0.264|
>
> **E2:** Irrelevant dependency.
>
> **R2:** "Irrelevant dependencies" refer to transient interactions caused by non-stationary patterns (extreme values, missing records, or noise) that lead to unstable statistical associations. As shown in https://i.postimg.cc/Jn42zB0b/rebuttal-fig.png, TimeFilter captures finer dependencies, distinguishing irrelevant ones. In contrast, PatchTST focuses on mutation points, and iTransformer spreads attention too broadly. This evidence shows how unfiltered dependencies propagate irrelevant ones, while our filtering mechanism preserves crucial context-aware relationships.
>
> ## Methods And Evaluation Criteria
> **M1:** Discussions exploring real-world scenarios.
>
> **R1:** We provide real-world examples based on intuition: user behavior with low inter-channel correlation benefits from tem. dependencies, while spa. dependencies are key for real-time monitoring. In traffic flow prediction, spa.-tem. dependencies capture dynamic interactions across locations. For more details, refer to Section 4.2 (Routing Network, lines 205-233).
>
> ## Weaknesses
> **W1:** Parameter Sensitivity.
>
> **R1:** The number of nodes $n$ is set as $n = C \times \lceil L/P \rceil$, where $P$ is the patch length. The result in table below shows that performance varies with $P$, as different patch lengths capture distinct temporal semantics based on the dataset's characteristics. The threshold $p$ influences the number of favored dependencies. Except for $p = 1.0$ (full retention), the model already removes irrelevant dependencies, making it less sensitive to $p$.
>
> ||TimeFilter($p=0.5$)|$P=96$|$P=48$|$P=32$|$P=16$|$p=0.3$|$p=0.9$|
> |-|-|-|-|-|-|-|-|
> |Weather|**0.239**/**0.269**|0.242/0.271|**0.239**/**0.269**|0.243/0.272|0.245/0.275|0.241/0.272|0.240/0.270|
> |ECL|**0.158**/**0.256**|0.163/0.259|0.166/0.263|**0.158**/**0.256**|0.160/0.258|0.162/0.258|0.159/0.257|
> |Traffic|**0.407**/**0.268**|**0.407**/**0.268**|0.427/0.278|0.433/0.283|0.430/0.281|0.415/0.276|0.413/0.272|
>
> **W2:** Paradigm Transformation.
>
> **R2:** Thank you for recognizing the novelty of our work. We’ll change "Paradigm Transformation" to "Novel Paradigm" and acknowledge prior channel strategies and GNNs.
>
> ## Questions
> **Q1&Q3:** Complexity.
>
> **R1:** We compare the theoretical complexity of TimeFilter with other Transformer-based models in the table below. $C$ is the number of channels, $L$ is the input length, and $P$ is the patch length.
> |TimeFilter|iTransformer|PatchTST|Crossformer|
> |-|-|-|-|
> |$O(C^2·(\frac{L}{P})^2)$|$O(C^2)$|$O(C·(\frac{L}{P})^2)$|$O(\frac{C}{P^2}·L^2)$|
>
> The graph construction and filtering modules have complexity $O((\frac{CL}{P})^2)$, and the GNN module is $O(CL)$. However, theoretical complexity alone doesn't fully capture real-world performance, as shown by testing on a NVIDIA A100 GPU with training (1 epoch) and inference times averaged over 5 runs:
> |||TimeFilter|iTransformer|PatchTST|Crossformer|
> |-|-|-|-|-|-|
> |Weather ($C=21$)|Training|19s|23s|41s|74s|
> ||Inference|4s|6s|9s|11s|
> |Electricity ($C=321$)|Training|91s|74s|280s|248s|
> ||Inference|14s|11s|36s|34s|
> |Traffic ($C=862$)|Training|77s|101s|352s|360s|
> ||Inference|14s|18s|66s|81s|
>
> Despite higher theoretical complexity, TimeFilter outperforms PatchTST in training and inference speed. This efficiency gain likely comes from structured sparsity in filtering, where einsum-based operations produce sparse matrices, reducing FLOPs and memory overhead.
>
> **Q2:** Is $m=2$ sufficient for modeling?
>
> **R2:** The value of $m$ is not a fixed hyperparameter. It represents the number of filter types, determined dynamically by the confidence threshold ($p$). Typically, one or two filters exceed $p$, so the dynamic routing mechanism selects one or two dependencies. When all three filters are active, it indicates a need for full channel dependency structures, though only a few patches require all dependencies.
>
> Thank you again for your careful review and constructive suggestions, which have inspired us to improve our paper further.

---

### Official Review · Reviewer_1FL4 · 2025-03-17

**Overall Recommendation:** 2

**Summary:**

In this paper, the authors propose to imporve multivariate time series (MTS) forecasting by proposing the TimeFilter framework, which introduces patch-specific spatial-temporal graph filtration to model dynamic dependencies. Traditional MTS forecasting approaches either follow a channel-independent (CI) approach, ignoring inter-channel dependencies, or a channel-dependent (CD) approach, which captures all dependencies indiscriminately, often leading to noise and reduced robustness. The authors argue that both methods have limitations, particularly in capturing time-varying dependencies. To address this, TimeFilter employs a fine-grained dependency modeling technique that filters out irrelevant correlations and retains only the most significant ones in a patch-specific manner. The framework is based on a graph neural network (GNN) architecture and dynamically adjusts its approach to each dataset's characteristics, using a mixture of experts mechanism. Extensive experiments across several real-world datasets demonstrate that TimeFilter outperforms state-of-the-art methods in both long- and short-term forecasting tasks.

**Claims And Evidence:**

The claims made in the submission are supported by clear and convincing evidence.

**Essential References Not Discussed:**

Please refer to "Methods And Evaluation Criteria" and "Experimental Designs Or Analyses"

**Experimental Designs Or Analyses:**

The proposed method is very similar to [4], but the authors do not consider it as baseline.

[4] Zhao, L. and Shen, Y. Rethinking channel dependence for multivariate time series forecasting: Learning from leading indicators. In The Twelfth International Conference on Learning Representations, 2024.

**Methods And Evaluation Criteria:**

One of the main concern is the limited contribution of this paper.  The proposed idea is conceptually similar to existing Granger causality-based methods [1, 2] and instantaneous time series [3], yet the authors fail to discuss this connection. Specificially, the temporal dependencies and spatial-temporal dependencies can be described by Granger Causality, and the inter-channel (spatial) dependencies can be described as instantaneous dependencies.


[1] Marcinkevičs, Ričards, and Julia E. Vogt. "Interpretable models for granger causality using self-explaining neural networks." arXiv preprint arXiv:2101.07600 (2021).
[2] Tank, Alex, et al. "Neural granger causality." IEEE Transactions on Pattern Analysis and Machine Intelligence 44.8 (2021): 4267-4279.
[3] Lippe, Phillip, et al. "Causal representation learning for instantaneous and temporal effects in interactive systems." arXiv preprint arXiv:2206.06169 (2022).

**Other Comments Or Suggestions:**

N.A.

**Other Strengths And Weaknesses:**

N.A.

**Questions For Authors:**

N.A.

**Relation To Broader Scientific Literature:**

N.A

**Theoretical Claims:**

The paper lacks theoretical grounding or in-depth analysis to explain why the proposed method leads to performance improvements. A deeper understanding of the underlying mechanisms would clarify and strengthen the contribution.

---

> ### Author Rebuttal · Authors · 2025-04-01
>
> Thanks for your valuable comments. Here are responses to your insightful concerns and questions:
>
> ---
> **`Q1`: Novelty and Contribution. (The proposed idea is conceptually similar to existing Granger causality-based methods [1, 2] and instantaneous time series [3], yet the authors fail to discuss this connection.)**
>
> **`R1`:** We would like to clarify that our core innovation lies in proposing a novel fine-grained dependency modeling paradigm. Existing methods predominantly focus on channel-wise correlations at a coarse granularity, whereas our approach explicitly models time-varying fine-grained dependencies that cannot be effectively captured by channel-wise interactions alone.
>
>   The foundation motivation of TimeFilter stems from the following observation:
>   Real-world time series often exhibit hybrid characteristics containing both causal signals (spa.-tem.&spa.) and inertial signals (tem.). However, the global time series signals do not conform to the causal, auto-generative nature assumed in modeling, but are influenced by unobserved factors [5]. Such forecasting are similar to the natural language understanding (NLU), where integrating all latent signals is more crucial. As the reviewer pointed out, Granger causality[1,2] only accounts for temporal inertia and spatio-temporal causality, while the instantaneous effects[3] only capture spatial causality. In contrast, TimeFilter integrates all potential signals during graph construction and then adaptively filters out invalid signals under specific segments, achieving superior forecasting capability.
>
>   To better demonstrate our novelty, we provide a comprehensive comparison across critical dimensions in the following table. (C channels, L lookback horizon, D hidden dimensions and K lead-lag steps).
> ||TimeFilter|Granger causality|Instantaneous effects|
> |-|-|-|-|
> |Signals|Tem.&Spa.&Spa.-Tem.|Spa.-Tem.&Tem.|Spa.|
> |Granularity|Patch-wise|Channel-wise|Channel-wise|
> |Assumption|None.|Static|Synchronization|
> |Robustness|Yes|No|No|
> |Complexity|$O(C^2(\frac{L}{P})^2)$|$O(C^2K+LCD+LD^2)$|$O(LC^3)$|
>
> [5] Time Series Prediction: Forecasting The Future And Understanding The Past
>
> [6] MTEB: Massive Text Embedding Benchmark
>
> ---
> **`Q2`: Analysis of why TimeFilter improve the performance. (The paper lacks theoretical grounding or in-depth analysis to explain why TimeFilter leads to performance improvements.)**
>
> **`R2`:**  We appreciate this insightful question regarding the validation of the effectiveness of our method. The core mechanism can be explained from two complementary perspectives:
>
> - **Theoretical foundation**: As noted in Q1, our filtering mechanism addresses the spurious regression artefacts prevalent in dependency modeling. Specifically, while some signals provide genuine predictive patterns, others exhibit localized spurious correlations arising from short-term segment analysis - a phenomenon particularly pronounced in non-stationary time series [7]. TimeFilter's dynamic pruning acts as a variance reduction operator, suppressing these transient noise signals while preserving true causal relationships.
>
> - **Empirical verification**: As shown in https://i.postimg.cc/Jn42zB0b/rebuttal-fig.png, we quantitatively demonstrate that in the Weather dataset. The filtered adjacency matrix exhibits clearer cluster structures aligned with physical sensor relationships. This fine-grained opeartion allows it to focus on high-spatial and high-temporal-spatial correlations (e.g., Fea0-Fea1 and Fea1-Fea3) while eliminating irrelevant dependencies (e.g., Fea0-Fea3).
>
> This dual validation by both causal theory and data-driven evidence confirms that our adaptive filtering successfully disentangles persistent patterns from transient noise, directly contributing to the observed performance gains.
>
> [7] Spurious Correlations in High Dimensional Regression: The Roles of Regularization, Simplicity Bias and Over-Parameterization.
>
> ---
> **`Q3`: Compare with LIFT[4]. (The proposed method is very similar to [4], but does not consider it as a baseline.)**
>
> **`R3`:** LIFT[4] is added into the baseline. Our method still outperforms it (PatchTST+LIFT) and we will include LIFT in the revised version. While LIFT models lead-lag dependency modeling through lead estimation and refinement, it is sensitive to data distribution and less effective for synchronous data without significant lead-lag relationships. TimeFilter enhances such generalization ability and customizes dependency types for datasets from different domains.
> ||TimeFilter|LIFT|
> |-|-|-|
> |Metric|MSE/MAE|MSE/MAE|
> |Weather|0.216/0.258|0.229/0.262|
> |ECL|0.150/0.246|0.158/0.252|
> |Traffic|0.360/0.254|0.386/0.260|
>
> ---
> Overall, thanks again for your valuable comments. We hope the detailed experiments and clarifications provided above address your concerns. If you have any further questions, please feel free to reach out, and we would be happy to provide additional clarifications.

---

### Decision · Program_Chairs · 2025-05-01

**Decision:**

Accept (poster)

**Comment:**

This paper introduces TimeFilter, a novel patch-wise dependency modeling framework that unifies and advances beyond CI, CD, and CC strategies for time series forecasting. By constructing spatial-temporal graphs with k-NN and using a mixture-of-experts mechanism, TimeFilter dynamically filters relevant dependencies at fine granularity. Extensive experiments and ablations demonstrate good performance against multiple baselines. The rebuttal addresses most of the reviewers' concerns. The authors are encouraged to incorporate those revisions (e.g., softening novelty claims, adding complexity analysis, and adding discussion on hyperparameter tuning).